# AXL confers intrinsic resistance to osimertinib and advances the emergence of tolerant cells

Hirokazu Taniguchi [1,2], Tadaaki Yamada [1,3], Rong Wang[1], Keiko Tanimura[3], Yuta Adachi[1],
Akihiro Nishiyama[1], Azusa Tanimoto[1], Shinji Takeuchi[1], Luiz H. Araujo[4], Mariana Boroni[5],
Akihiro Yoshimura[3], Shinsuke Shiotsu[6], Isao Matsumoto[7], Satoshi Watanabe[8], Toshiaki Kikuchi[8],
Satoru Miura[9], Hiroshi Tanaka[9], Takeshi Kitazaki[10], Hiroyuki Yamaguchi[2], Hiroshi Mukae[2],
Junji Uchino[3], Hisanori Uehara[11], Koichi Takayama[3] & Seiji Yano [1,12]

A novel EGFR-tyrosine kinase inhibitor (TKI), osimertinib, has marked efficacy in patients with *EGFR*-mutated lung cancer. However, some patients show intrinsic resistance and an insufficient response to osimertinib. This study showed that osimertinib stimulated AXL by inhibiting a negative feedback loop. Activated AXL was associated with EGFR and HER3 in maintaining cell survival and inducing the emergence of cells tolerant to osimertinib. AXL inhibition reduced the viability of EGFR-mutated lung cancer cells overexpressing AXL that were exposed to osimertinib. The addition of an AXL inhibitor during either the initial or tolerant phases reduced tumor size and delayed tumor re-growth compared to osimertinib alone. AXL was highly expressed in clinical specimens of EGFR-mutated lung cancers and its high expression was associated with a low response rate to EGFR-TKI. These results indicated pivotal roles for AXL and its inhibition in the intrinsic resistance to osimertinib and the emergence of osimertinib-tolerant cells.

[1] Division of Medical Oncology, Cancer Research Institute, Kanazawa University, 13-1, Takara-machi, Kanazawa 920-0934, Japan. [2] Department of Respiratory Medicine, Nagasaki University Graduate School of Biomedical Sciences, 1-7-1 Sakamoto, Nagasaki 852-8501, Japan. [3] Department of Pulmonary Medicine, Graduate School of Medical Science, Kyoto Prefectural University of Medicine, 465, Kajii-cho, Kamigyo-ku, Kyoto 602-8566, Japan. [4] Division of Clinical Research, Brazilian National Cancer Institute, Rua André Cavalcanti 37, Rio de Janeiro — RJ 20231-050, Brazil. [5] Division of Experimental and Translational Research, Bioinformatics and Computational Biology Lab, Brazilian National Cancer Institute, Rua André Cavalcanti 37, Rio de Janeiro — RJ 20231-050, Brazil. [6] Department of Respiratory Medicine, Japanese Red Cross Kyoto Daiichi Hospital, 15-749 Hon-machi, Higashiyama-ku, Kyoto 605-0981, Japan. [7] Department of Thoracic, Cardiovascular and General Surgery, Kanazawa University, 13-1, Takara-machi, Kanazawa 920-0934, Japan. [8] Department of Respiratory Medicine and Infectious Diseases, Niigata University Graduate School of Medical and Dental Sciences, 1–757 Asahimachi-dori, Chuo-ku, Niigata 951-8510, Japan. [9] Department of Internal Medicine, Niigata Cancer Center Hospital, 2-15-3 Kawagishi-cho, Niigata 961-8566, Japan. [10] Department of Respiratory Medicine, Japanese Red Cross Nagasaki Genbaku Hospital, 3-15 Mori-machi, Nagasaki 852-8511, Japan. [11] Department of Pathology and Laboratory Medicine, Institute of Biomedical Sciences, Tokushima University Graduate School, 3-18-15, Kuramoto-cho, Tokushima 770-8503, Japan. [12] Nano Life Science Institute, Kanazawa University, Kakuma, Kanazawa 920-1192, Japan. These authors contributed equally: Hirokazu Taniguchi, Tadaaki Yamada. Correspondence and requests for materials should be addressed to T.Y. (email: tayamada@koto.kpu-m.ac.jp) or to S.Y. (email: syano@staff.kanazawa-u.ac.jp)

Non-small cell lung cancer (NSCLC) with activating mutations in the epidermal growth factor receptor (EGFR) such as an exon 19 deletion and L858R mutation responds to first- and second-generation EGFR-tyrosine kinase inhibitors (EGFR-TKIs), including gefitinib, erlotinib, and afatinib[1–4]. However, 20–30% of NSCLC patients with mutated EGFR are insensitive to EGFR-TKIs and are clinically nonresponders. Several factors regarding the mechanisms of the intrinsic resistance have been previously reported including the EGFR-T790M mutation, EGFR-exon20 insertions, overexpression of hepatocyte growth factor (HGF), and BIM deletion polymorphism[5–7]. In addition, about 30% of responders experience early relapse within 6 months due to acquired resistance[8]. The gatekeeper mutation EGFR-T790M is the most common cause of acquired resistance and is detectable in approximately 50% of patients who experience acquired resistance to first- and second-generation EGFR-TKIs[9,10].

Osimertinib is a third-generation EGFR-TKI and inhibits EGFR that have activating mutations and/or the T790M resistance mutation, but not wild-type EGFR or other kinases, such as AXL, AKT1, or HER3; thus, it is called the mutant-EGFR-specific inhibitor[11]. It is approved for the treatment of EGFR-T790M-positive NSCLC patients who acquired resistance to first-generation or second-generation EGFR-TKIs. These tumors are known to acquire resistance to osimertinib by mechanisms that include acquisition of the EGFR-C797S mutation, loss of the T790M mutation, activation of a bypass pathway, and histological transformation including small cell transformation[12]. Much attention has been paid to the efficacy of osimertinib in the first-line setting of clinical intervention. A recent phase III clinical trial (FLAURA) demonstrated that in patients with EGFR-mutated NSCLC, progression-free survival (PFS) was longer for those treated with first-line osimertinib than for those treated with gefitinib or erlotinib[13]. Based on the results of the FLAURA study, first-line treatment with osimertinib is considered as one of the standard treatments for NSCLC patients with mutated EGFR. However, it was also noted that a population of patients showed intrinsic resistance and an insufficient response to osimertinib treatment, similar to that observed with other EGFR-TKIs. Moreover, even in responders, the rate of complete response is very low (3%) and residual lesions remain in the majority of the patients. Such residual lesions may serve as the basis of recurrent disease. Therefore, an understanding of the molecular mechanisms underlying intrinsic resistance and early refractoriness to osimertinib in EGFR-TKI naïve EGFR-mutated NSCLC is needed in order to establish novel therapies.

A recently identified small subpopulation of reversibly drug-tolerant (DT) cells with >100-fold reduced drug sensitivity has been reported to maintain viability via IGF-1 receptor signaling[14] and is supposed to contain residual lesions in mutated EGFR of NSCLC treated with first- and second-generation EGFR-TKIs. However, the molecular mechanism by which NSCLC cells emerge as DT cells to osimertinib is totally unknown.

AXL is the receptor for tyrosine kinase and was first identified in 1991 in two patients with chronic myeloid leukemia[15]. High expression of the AXL protein in tumors is reported to be associated with poor prognosis in patients with several types of cancer including glioblastoma, breast cancer, lung cancer, and acute myeloid leukemia[16–19]. Overexpression of AXL has been detected more frequently in lung adenocarcinomas that harbor EGFR-activating mutations, compared with NSCLC that have wild-type EGFR[20]. Moreover, investigators from several studies have reported that the activation of AXL signaling in tumors is associated with acquired resistance to several targeted molecular therapy drugs and chemotherapeutic agents. However, its roles in intrinsic resistance and DT cells regarding specific targeted molecular therapy drugs remain unknown[21–24].

The current study was designed to clarify the molecular mechanisms underlying the induction of tolerance to osimertinib in EGFR-mutated NSCLC cells, including the roles of AXL signaling during the initial and tolerant phases. We determined that AXL interacted with other molecules, including EGFR and HER3, and maintained survival of tumor cells exposed to osimertinib. AXL inhibition using specific siRNA or chemical compounds in tumor cell-derived xenograft (CDX) models and patient-derived xenograft (PDX) models reduced the viability of osimertinib-exposed tumor cells, inhibited the emergence of DT cells, and delayed the recurrence of tumors.

## Results

**Osimertinib activated AXL and re-activated HER3 and EGFR.** EGFR-mutated NSCLC cells, including PC-9 cells, were highly sensitive to the third-generation EGFR-TKI osimertinib. As the $C_{max}$ in patient plasma following oral administration of 80 mg of osimertinib was 635.4 nM[25], we considered that 1 μM or less would be a clinically relevant concentration of osimertinib. However, approximately 20–30% of the PC-9 cells survived, even after treatment with a high concentration (1 μM) of osimertinib for 72 h (Supplementary Figure 1). To determine the mechanism by which these cells escaped the effects of osimertinib, we evaluated 43 phospho-kinases in PC-9 cells treated with or without osimertinib for 72 h using a phospho-kinase antibody array. We found that the phosphorylation of some molecules, such as HER3, MET, and AXL, increased after osimertinib treatment (Fig. 1a). We therefore hypothesized that tolerance to osimertinib may be caused by a survival signal via these proteins. Kinetic analysis revealed that while phosphorylation of EGFR was remarkably inhibited by osimertinib at 4 h, it was measurably re-activated at 72 h (Fig. 1b). The phosphorylation of HER3 and MET was also inhibited by osimertinib at 4 h, but the phosphorylation of these proteins was re-activated at 24 h and the reactivation gradually increased through 72 h. On the other hand, while AXL was not constitutively phosphorylated, its phosphorylation was induced by osimertinib at 4 h and increased through 72 h. Phosphorylation of ERK, a downstream molecule of EGFR, was remarkably inhibited by osimertinib at 4 h and the suppression was maintained through 72 h. In sharp contrast, AKT, another downstream molecule of EGFR signaling, was slightly inhibited by osimertinib at 4 h and was re-activated at 24 h. These results suggested that osimertinib exposure may have activated AXL and thereby re-activated HER3, MET, and EGFR in PC-9 cells.

We next examined the effect of knockdown of HER3, MET, and AXL on the viability of PC-9 and PC-9GXR cells, which have exon 19 deleted and the T790M mutation in EGFR. In the absence of osimertinib, knockdown of HER3, MET, and AXL using specific siRNAs resulted in the inhibition of PC-9 and PC-9GXR cell viability by 30–40%, 25%, and less than 20%, respectively (Fig. 1c). Osimertinib inhibited the viability of both PC-9 and T790M-positive PC-9GXR cells by 50%, consistent with its activity as third-generation EGFR-TKI. In the presence of osimertinib for 72 h, knockdown of MET did not affect cell viability, while knockdown of HER3 or AXL further decreased the viability of PC-9 and PC-9GXR cells to about 20%. These results suggested that AXL and HER3 may have promoted the survival of a subset of EGFR-mutated NSCLC cells in which the EGFR signal was inhibited by the 72-h exposure to osimertinib. In comparison to osimertinib treatment, when PC-9 and PC-9GXR cells were treated with siRNA specific for EGFR, cell viability was reduced by 25–30%. Knockdown of HER3 also reduced cell viability by 25–30%, but knockdown of AXL

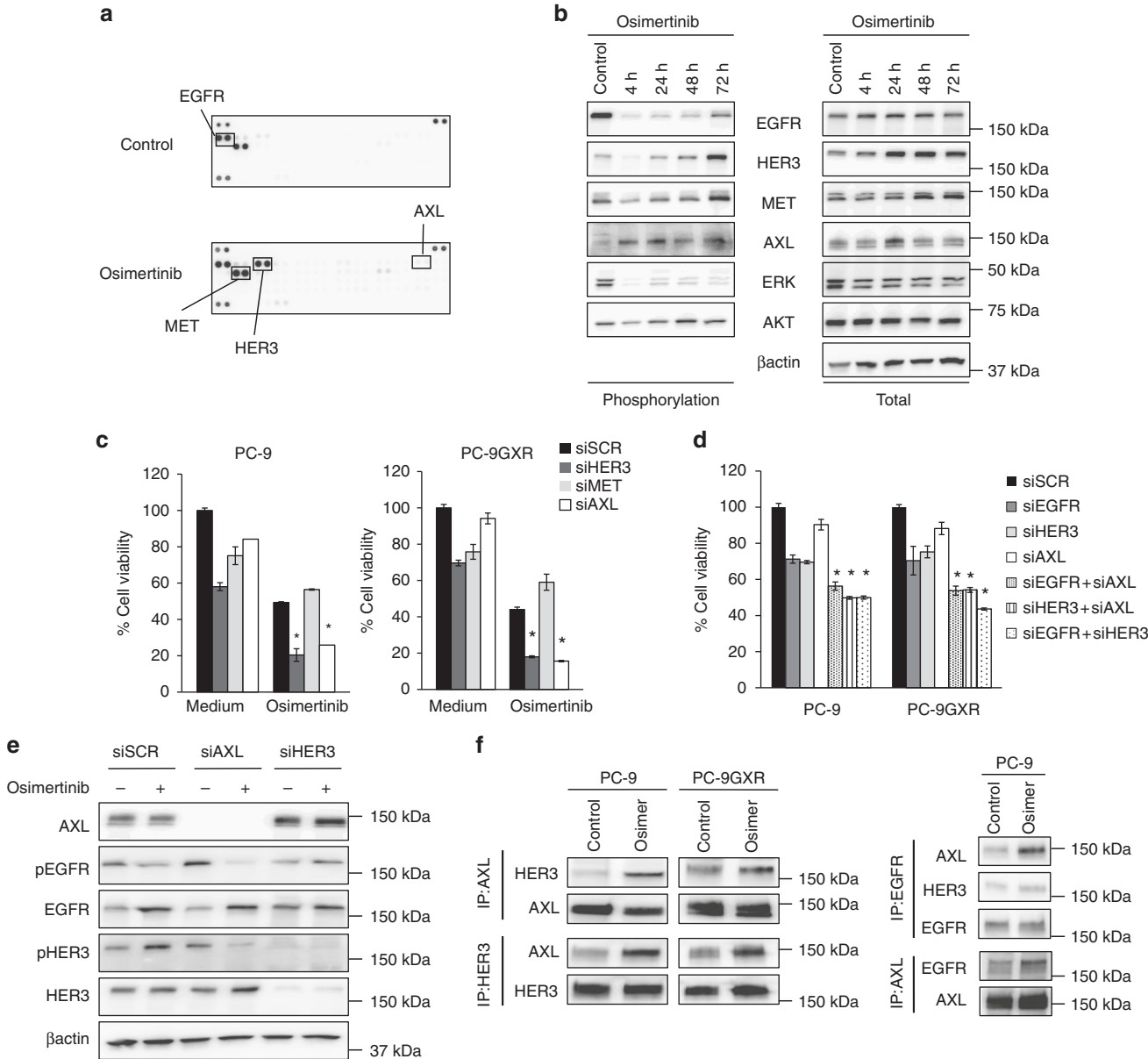

**Fig. 1** Osimertinib activated AXL in *EGFR*-mutated NSCLC cells in vitro. **a** Human tyrosine kinase phosphorylation array analysis of PC-9 cells in the presence or absence of osimertinib (100 nmol/L) for 72 h. **b** PC-9 cells were treated with osimertinib (100 nmol/L), lysed, and the indicated proteins detected by western blotting. **c** Nonspecific siRNA control, specific siRNA for *AXL*, *HER3* or *MET* introduced into the indicated cells. After 24 h, the cells were incubated with or without osimertinib (100 nmol/L) for 72 h and cell viability was determined using MTT assays. *$P < 0.05$ for comparisons of osimertinib-treated cells with parental cells treated with nonspecific control siRNA. Comparisons by paired Student's *t* tests. **d** PC-9 cells were treated for 72 h with the indicated siRNAs, or combinations of the indicated siRNAs and cell viability was determined using MTT assays. *$P < 0.05$ compared with the respective parental cells. Comparisons by paired Student's *t* tests. **e** The indicated siRNAs were introduced into PC-9 cells. After 24 h, the cells were incubated with or without osimertinib (100 nmol/L) for 72 h and lysed, and the indicated proteins detected by western blotting. **f** Cell lines were treated with or without osimertinib (100 nmol/L) for 72 h. The cells were lysed and the indicated proteins were detected by western blotting with immunoprecipitation of the indicated proteins

only marginally reduced cell viability. These results are consistent with previous findings that heterodimerization of EGFR and HER3 contributes to the maintenance of oncogenic signaling in *EGFR*-mutated NSCLC cells[26]. Under these experimental conditions, dual knockdown of *EGFR* and either *AXL* or *HER3* showed greater reductions in cell viability compared with the knockdown of *EGFR* alone (Fig. 1d). Interestingly, dual knockdown of *HER3* and *AXL* decreased cell viability as effectively as the dual knockdown of *EGFR* and *AXL*, suggesting an interaction between AXL and EGFR or HER3.

We next examined the phosphorylation status of these molecules in PC-9 cells, with or without the knockdown of *AXL* or *HER3*. Treatment of PC-9 cells with specific siRNAs efficiently knocked down the protein expression of either AXL or HER3 (Fig. 1e). In PC-9 cells treated with control siRNA, 72 h exposure to osimertinib resulted in only slight suppression of EGFR phosphorylation, consistent with the results shown in Fig. 1b, presumably due to reactivation. Interestingly, in PC-9 cells treated with siRNA for *AXL*, the same 72 h osimertinib treatment greatly inhibited phosphorylation of EGFR and HER3.

This phenomenon was not observed in PC-9 cells treated with siRNA for *HER3*. Immunoblots after immunoprecipitation revealed that 72 h treatment with osimertinib enhanced the binding of AXL to HER3 and to EGFR, but not the binding of to HER3 to EGFR (Fig. 1f). These results indicated that AXL was activated by osimertinib exposure and maintained cell survival in some of the PC-9 and PC-9GXR cells, presumably by interacting with EGFR and HER3.

To assess the mechanism by which AXL phosphorylation was adversely activated by osimertinib, we investigated the involvement of the negative feedback loop from ERK to the sprouty (SPRY) family proteins, which inhibits phosphorylation of various receptor tyrosine kinases[27,28]. Osimertinib treatment decreased the levels of SPRY4 in PC-9 cells (Supplementary Figure 2A). Knockdown of *SPRY4* using specific siRNA increased the expression of phosphorylated AXL (Supplementary Figure 2B). In contrast, overexpression of SPRY4 maintained expression levels of phosphorylated AXL in PC-9 cells exposed to osimertinib (Supplementary Figure 2C). These results indicated that osimertinib adversely activated AXL, at least in part, by shutting off the negative feedback loop to SPRY4, which suppressed AXL phosphorylation (Supplementary Figure 2D).

**AXL inversely correlated with susceptibility to EGFR-TKIs**. We next sought to evaluate the correlation between AXL expression and susceptibility to EGFR-TKIs, including osimertinib, in *EGFR*-mutated NSCLC cell lines. The nine tested *EGFR*-mutated NSCLC cell lines were categorized into those with high levels of AXL expression (AXL-βactin ratio of >7.0) and those with low levels of AXL expression (AXL-βactin ratio of ≦7.0). EGFR phosphorylation and SPRY4 levels tended to be higher in the cells with the relatively lower AXL expression levels (Spearman rank correlation = −0.6 and $P = 0.048$; Spearman rank correlation = −0.644 and $P = 0.061$, relatively); whereas, there were no clear differences in the expression of epithelial−mesenchymal transition (EMT)-related proteins, total EGFR, or phosphorylated HER3 (Fig. 2a and Supplementary Figures 3A, 3B, 3C, and 4). Importantly, the half-maximal inhibitory concentrations (IC50s) of both osimertinib and gefitinib were significantly higher in cells demonstrating the higher relative AXL expression levels than in the lower expressing cells (Fig. 2b). Moreover, the AXL expression levels had a most strong positive correlation with osimertinib susceptibility among several related proteins (Spearman rank correlation = 0.733; $P = 0.031$) (Supplementary Figure 5, 6, supplementary Table 1).

We also retrospectively assessed the correlations between AXL expression and the clinical efficacy of the first- and second-generation EGFR-TKIs and osimertinib in 46 patients with *EGFR*-mutated NSCLC and in 11 patients with *EGFR*-mutated NSCLC harboring the T790M mutations, respectively. Expression of AXL in the cell cytoplasm of pre-EGFR-TKI-treated tumor samples was evaluated using immunohistochemistry (IHC) staining and scored as high (3+), intermediate (2+), low (1+), and no (0) expression of AXL (Supplementary Figure 7A). Of the 46 *EGFR*-mutated NSCLC tumor specimens for the first- and second-generation EGFR-TKIs, high, intermediate, low, and no AXL expression was observed in 12 (26.1%), eight (17.4%), 23 (50.0%), and three (6.5%) specimens, respectively. Of the 11 *EGFR*-mutated NSCLC tumors for osimertinib, high, intermediate, low, and no AXL expression was observed in three (27.2%), one (18.2%), seven (63.6%), and zero (0.0%) specimens, respectively. The response rate for the first- and second-generation EGFR-TKIs for the patients with AXL expression scores of 0 to 2+ was high (87.5−100%), whereas for those patients with AXL expression scores of 3+, the response rate for

the first- and second-generation EGFR-TKIs was significantly lower (50%) ($P < 0.001$) (Fig. 2c). The response rate for osimertinib for the patients with AXL expression scores of 0 to 2+ was high (85.7−100%), while for those patients with AXL expression scores of 3+, the response rate for osimertinib was relatively lower (66.7%) (Fig. 2d). Moreover, the PFS following treatment with the first- and second-generation EGFR-TKIs and osimertinib trended toward being shorter in the patients with AXL expression scores of 3+, compared with those with AXL expression scores of 0 to 2+ ($P = 0.168$ and hazard ratio (HR) = 0.58; $P = 0.449$ and HR = 0.81, respectively) (Supplementary Figure 7B, 7C).

These findings suggested that AXL expression correlated with a poor initial response to EGFR-TKI treatment, including osimertinib, and correlated to early relapse.

**AXL knockdown sensitized AXL-expressing cells to osimertinib**. We examined whether the knockdown of AXL protein expression increased the sensitivity of *EGFR*-mutated NSCLC cells to EGFR-TKIs, including osimertinib. Treatment with *AXL*-specific siRNA enhanced the inhibitory effects of osimertinib on the viability of cell lines PC-9, PC-9GXR, and HCC4011, all of which express high levels of AXL, but had only a marginal effect on the viability of cell lines HCC827, HCC4006, and H3255, which express low levels of AXL (Fig. 3a). Similar results were observed with treatment with gefitinib except in PC-9GXR cells, which have the *EGFR*-T790M mutation. The IC50 values for osimertinib and gefitinib in the high-AXL-expressing cell lines treated with anti-*AXL* siRNA were significantly lower than those treated with control siRNA ($P = 0.0014$ for osimertinib and $P = 0.0028$ for gefitinib) (Fig. 3b). Treatment with *AXL*-specific siRNA did not significantly affect IC50 values for osimertinib or gefitinib in low-AXL-expressing cells. Western blot analysis showed that treatment of PC-9, PC-9GXR, and HCC4011 cells with osimertinib plus *AXL*-specific siRNA inhibited the phosphorylation of AKT more than did treatment with osimertinib plus control siRNA (Fig. 3c). In contrast, osimertinib plus *AXL*-specific siRNA treatment of the cell lines did not affect the phosphorylation of ERK compared with osimertinib plus control siRNA treatment. These findings suggested that *AXL* knockdown resulting in the suppression of the AKT axis may have sensitized high-AXL-expressing *EGFR*-mutated NSCLC cells to EGFR-TKIs and reduced cell survival after 3 days of treatment.

**AXL inhibitor sensitized AXL-expressing cells to osimertinib**. We evaluated whether AXL inhibitors could increase the sensitivity of *EGFR*-mutated NSCLC cells to EGFR-TKIs. NPS1034, an AXL inhibitor, did not affect the viability of the *EGFR*-mutated NSCLC cell lines that were tested. The use of NSP1034 in combination with the third-generation EGFR-TKI osimertinib or rociletinib for 72 h increased cell sensitivity to these EGFR-TKIs and reduced the viability of high-AXL-expressing PC-9, PC-9GXR, and HCC4011 cells, but not that of the low-AXL-expressing HCC827 cells (Fig. 4a and Supplementary Figure 8). We proved that AXL was the target of NPS1034 for the sensitization of *EGFR*-mutated NSCLC cells to osimertinib using wild-type AXL overexpression system in PC-9 cells (Supplementary Figure 9A, 9B).

Consistent with the results of treating cells with *AXL*-specific siRNA combined with the use of osimertinib and NPS1034 for 1 h or 72 h more remarkably inhibited the phosphorylation of AXL, EGFR, HER3, and AKT compared with treatment of the high-AXL-expressing cell lines with osimertinib alone (Fig. 4b, c). In contrast, in the low-AXL-expressing cell lines, osimertinib and NPS1034 did not affect the phosphorylation of AKT, compared to

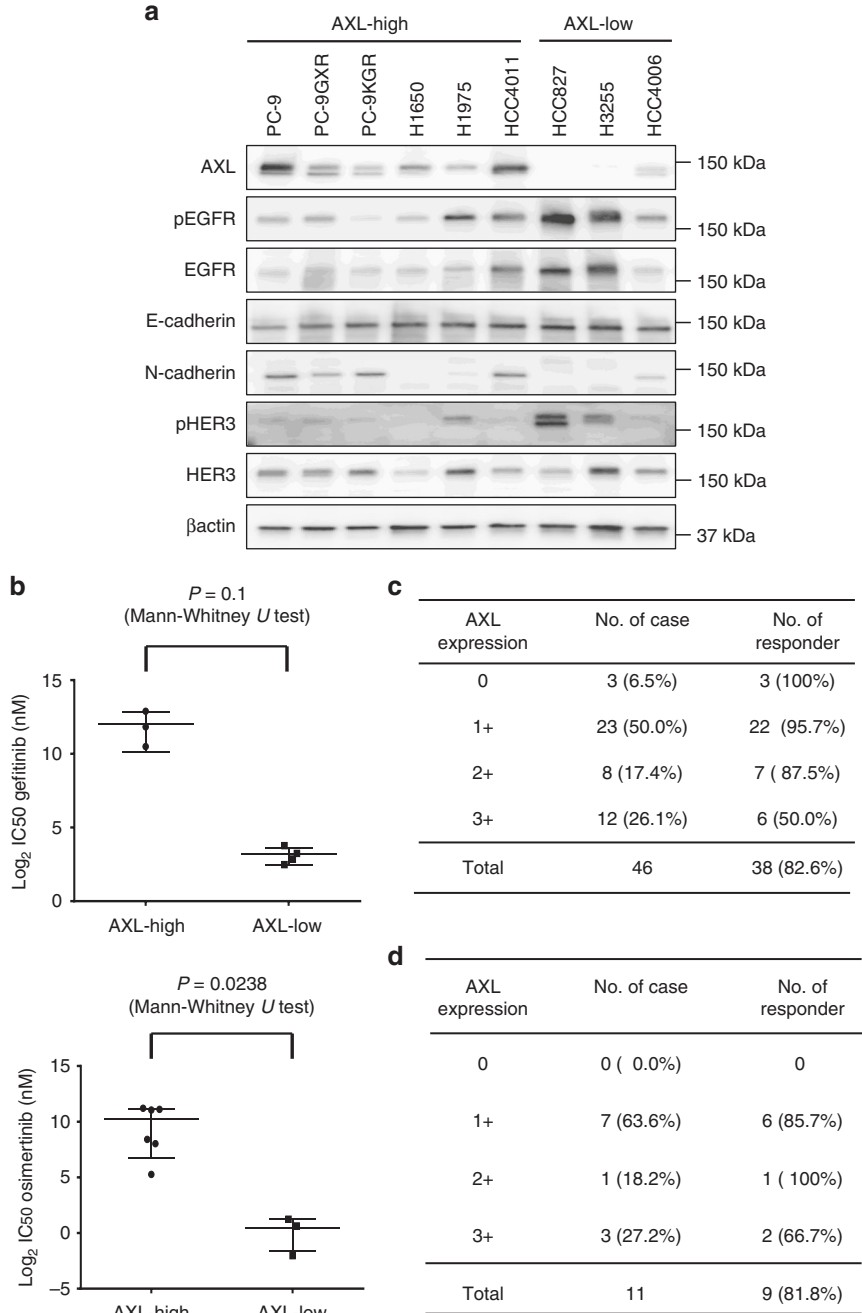

**Fig. 2** AXL protein expression inversely correlated with susceptibility to EGFR-TKIs. **a** The *EGFR*-mutated NSCLC cell lines PC-9, PC-9GXR, PC-9KGR, H1650, H1975, HCC4011, HCC827, H3255, and HCC4006 were lysed and the indicated proteins were detected by western blotting. **b** The IC50 values for the EGFR-TKIs gefitinib and osimertinib in *EGFR*-mutated NSCLC cells. The IC50 values for both gefitinib and osimertinib were significantly higher in cells expressing high levels of AXL compared to cells expressing low levels of AXL. *P* values were calculated using the Mann Whitney *U* test. **c** Correlation between the cytoplasmic AXL protein expression levels determined immunohistochemically and the response to treatment with EGFR-TKIs in *EGFR*-mutated NSCLC specimens from 46 patients. **d** Correlation between the expression levels of the cytoplasmic AXL protein, determined immunohistochemically, and response to treatment with osimertinib in *EGFR*-mutated NSCLC specimens from 11 patients

osimertinib alone (Supplementary Figure 10). Combined use of another AXL inhibitor (ASP2215) with osimertinib for 72 h also produced similar effects in PC-9 and HCC4011 cells (Supplementary Figure 11) compared with the treatment of the cells with osimertinib alone.

To further elucidate the profiles of the downstream molecules of the AXL axis, we investigated the combined efficacy of osimertinib with molecular-targeting drugs, such as those targeting AKT, MEK, or NFκB, in *EGFR*-mutated NSCLC cells

with a high level of AXL. Of them, the AKT inhibitor buparlisib showed additional effects when administered with osimertinib, compared to trametinib (an MEK inhibitor) or caffeic acid phenethyl ester (CAPE, an NFκB inhibitor) (Supplementary Figure 12). The expression of stem cell- and EMT-related proteins was not remarkably affected by osimertinib with or without NPS1034 (Supplementary Figure 13).

These results clearly indicated that the cell sensitivity to osimertinib could be increased by the combined treatment with

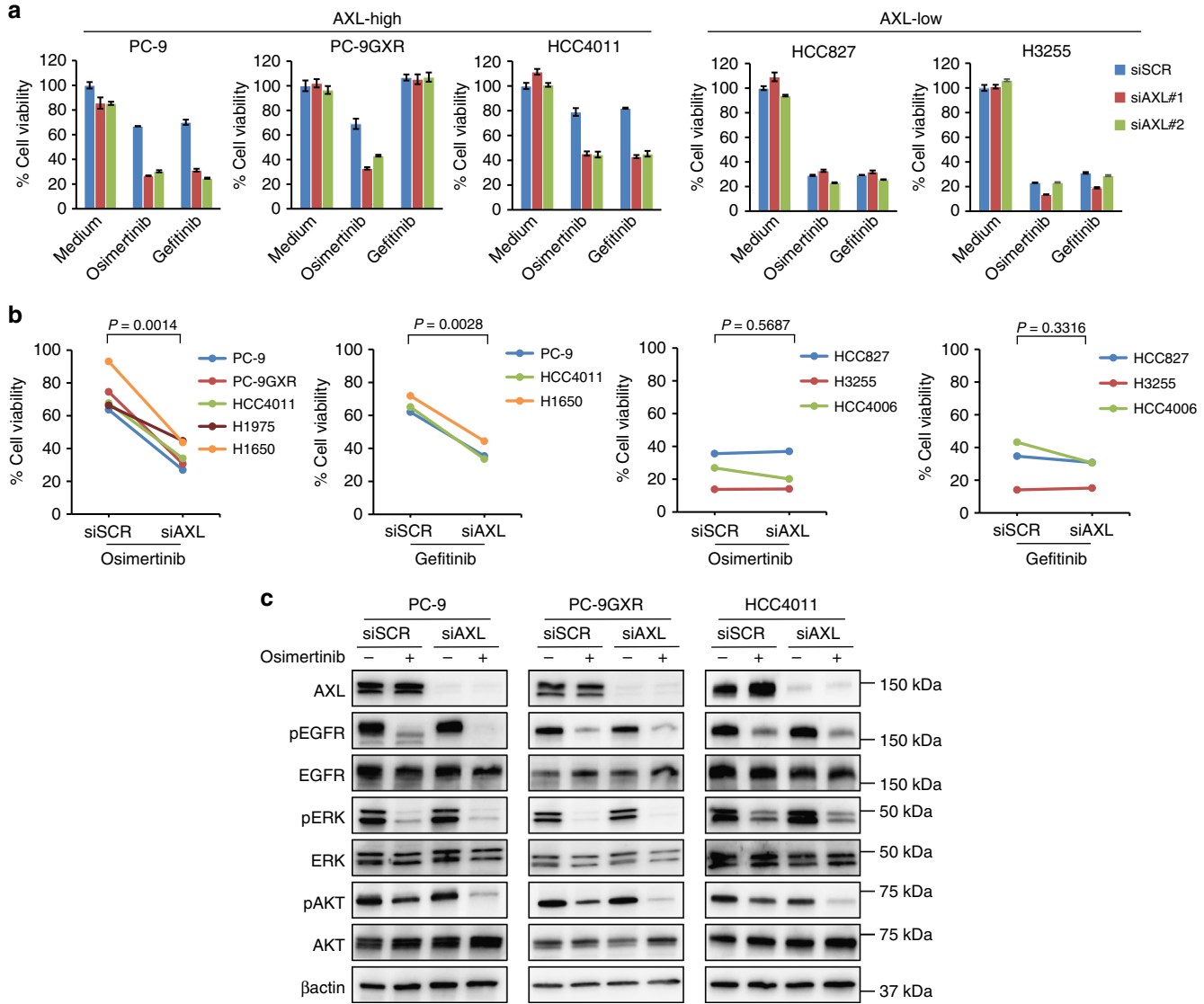

**Fig. 3** AXL knockdown sensitized *EGFR*-mutated NSCLC cells to osimertinib in vitro. **a** Nonspecific siRNA control or *AXL*-specific siRNAs (#1 and #2) were introduced into PC-9, PC-9GXR, HCC4011, HCC827, and H3255 cells. After 24 h, the cells were incubated with osimertinib (100 nmol/L) or gefitinib (100 nmol/L) for 72 h and the cell growth was determined using MTT assays. The percentage of growth is shown relative to the growth of untreated control cells. **b** Quantitative determination of the inhibition of cell viability of high-AXL-expressing and low-AXL-expressing *EGFR*-mutant cells transfected with nonspecific siRNA control or *AXL*-specific siRNAs after treatment with osimertinib or gefitinib. Paired Student's *t* tests were used for comparisons. **c** Nonspecific siRNA control or *AXL*-specific siRNAs were introduced into PC-9, PC-9GXR, and HCC4011 cells. After 24 h, the cells were incubated with osimertinib (100 nmol/L) for 1 h. The cells were lysed and the indicated proteins were detected by western blotting

an AXL inhibitor, by predominantly modulating the AKT activity, resulting in reduced viability of high-AXL-expressing *EGFR*-mutated NSCLC cell lines.

**AXL inhibitor prevented the emergence of drug-tolerant cells**. Drug-tolerant (DT) cells are defined as a small subpopulation of cells with remarkably reduced sensitivity to targeted drugs. They are generated within several days to several weeks of exposure to target drugs. DT cells are believed to be the basis for tumor recurrence due to the cells acquiring resistance. Thus, we examined whether AXL was involved in the emergence of DT cells exposed to osimertinib. We isolated DT cells after 9 days of exposure of PC-9 and HCC4011 cells to osimertinib (1–3 μM). A large population of PC-9 DT cells were in the G1 phase of the cell cycle, as previously reported[14] (Supplementary Figure 14). Although the percentage of mutated *EGFR*-allele and the copy

number of the *EGFR* gene were not affected in the DT cells (Supplementary Table 2), the DT cells were highly insensitive to osimertinib compared with their parental cells (Fig. 5a). A previous study demonstrated that DT cells derived from PC-9 cells exposed to erlotinib maintained their viability via IGF-1R signaling[14]. Consistent with this previous report, we found that the DT cells resistant to osimertinib had higher expression and phosphorylation levels of the IGF-1R protein compared with parental PC-9 cells (Fig. 5b). Moreover, the DT cells expressed higher levels of EGFR, HER3, and AXL compared with that in the parental cells (Fig. 5b). Interestingly, while AXL phosphorylation increased, the phosphorylation of EGFR and HER3 decreased in DT cells compared with that in parental cells, suggesting a dependency on AXL and IGF-1R for the viability of DT cells. In fact, more AXL protein was associated with EGFR and HER3 in the DT cells compared to that in the parental cells (Fig. 5c). Both

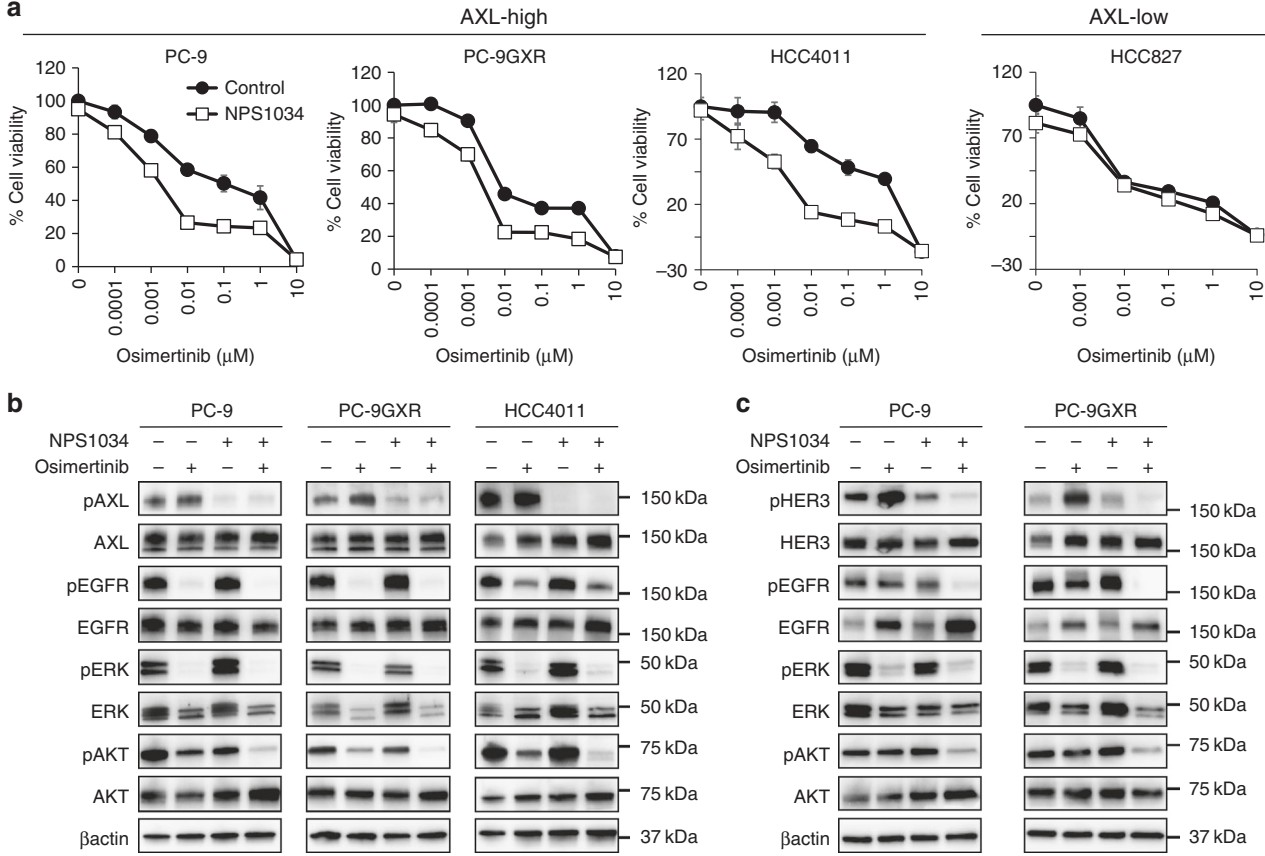

**Fig. 4** AXL inhibitor sensitized AXL-high *EGFR*-mutant NSCLC cells to osimertinib. **a** PC-9, PC-9GXR, HCC4011, and HCC827 cells were incubated with osimertinib in the presence or absence of AXL inhibitor NPS1034 (1 μmol/L) for 72 h and the cell viability was determined using MTT assays. Data are representative of three independent experiments that produced similar results. **b**, **c** The indicated cells were incubated with osimertinib (100 nmol/L) with or without NPS1034 (1 μmol/L) for 1 h (**b**) and 72 h (**c**). The cells were lysed and the indicated proteins were detected by western blotting

the AXL inhibitor (NPS1034) and IGF-1R inhibitor (OSI906) discernibly decreased the viability of DT cells, but not that of the parental PC-9 or HCC4011 cells (Fig. 5d). The combined treatment of DT cells with NPS1034 and OSI906 further inhibited their viability. Western blotting analysis showed that while osimertinib did not inhibit the phosphorylation of EGFR, HER3, ERK, or AXL in the DT cells, NSP1034 treatment alone inhibited the phosphorylation of AXL, EGFR, and HER3, and thus suppressed the phosphorylation of AKT but did not affect the phosphorylation of ERK (Fig. 5e). Moreover, the continuous cotreatment of PC-9, PC-9GXR, and HCC4011 cells with osimertinib and NPS1034 prevented the emergence of DT cells (Fig. 5f). These results indicated that in addition to IGF-1R, AXL played a pivotal role in the emergence of DT cells to osimertinib. Intriguingly, the AXL-mediated resistance was irreversible at least for 9 days with drug-free condition and was not easily reversible (Supplementary Figure 15).

**AXL inhibition with osimertinib shrunk in vivo tumors**. To determine whether AXL affected the sensitivity of high-AXL-expressing *EGFR*-mutated NSCLC cells to osimertinib in vivo, we used PC-9 cells in which AXL expression was continuously knocked down by transfection of short hairpin RNAs (shRNAs). Transfection of shRNA specific for *AXL* (#37 and #38) resulted in the inhibition of AXL protein expression with AXL shRNA #38 being more efficient than was AXL shRNA #37 at *AXL* knockdown (Fig. 6a). While the growth of PC-9 cells transfected with the shRNA was slightly slower than that of PC-9 cells transfected with a control shRNA (PC-9shSCR), both the knockdown

transfectants grew at a constant rate in vitro (Fig. 6b). Therefore, we chose the *AXL* shRNA #38-transfected PC-9 cells (PC-9shAXL#38) for use in the subsequent experiments. PC-9shAXL#38 cells were more sensitive to osimertinib and gefitinib than control PC-9shSCR cells (Fig. 6c). In a subcutaneous tumor model, the growth rates of tumors derived from PC-9shAXL#38 cells were slower than those derived from PC-9shSCR cells (Fig. 6d). In the model using PC-9shSCR cells, continuous osimertinib treatment initially resulted in slight tumor regression but the tumors re-grew within 10 days, indicating a rapid recurrence. In sharp contrast, the same treatment in the model using PC-9shAXL#38 cells resulted in dramatic tumor shrinkage with the tumors never re-growing in 24 days, the duration of the experiments. In the tumors derived from PC-9shAXL#38 cells, expression of the AXL protein was successfully knocked down (Fig. 6e). The number of Ki-67-positive proliferating tumor cells was significantly lower in the osimertinib-treated tumors derived from PC-9shAXL#38 cells than in those derived from PC-9shSCR cells (Fig. 6f, g). The C797S *EGFR*-mutation did not appear in each group of mice at the end of the experiment (Supplementary Table 3). None of the groups of mice showed a significant weight loss through the 24 days of treatment (Supplementary Figure 16A). These data clearly indicated that AXL retained the viability of osimertinib-treated PC-9 cells in vivo and that AXL knockdown improved the response of the tumors to osimertinib treatment and prevented recurrence.

**AXL inhibitor with osimertinib prevented CDX tumor regrowth**. We next evaluated the effect of NSP1034 treatment

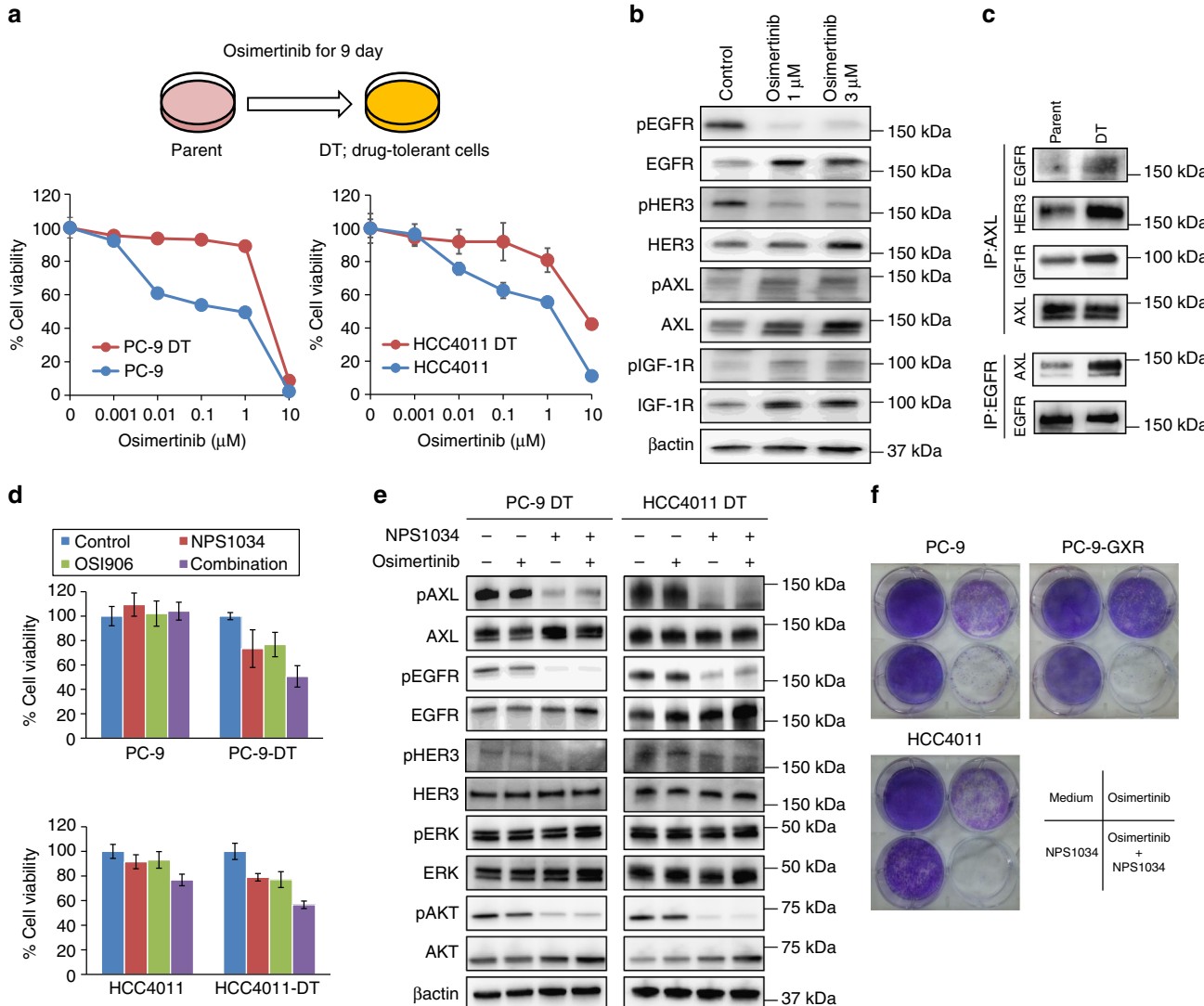

**Fig. 5** AXL inhibitor prevented the emergence of drug-tolerant cells to osimertinib. **a** Drug-tolerant (DT) cells previously treated with 3 μmol/L osimertinib for 9 days were treated with the indicated concentrations of osimertinib for 72 h and their viability was assessed using MTT assays. **b** PC-9 parental cells and DT cells generated by treatment with 1 or 3 μmol/L osimertinib were lysed and the indicated proteins were detected by western blotting. **c** PC-9 parental cells and DT cells generated by treatment with 3 μmol/L osimertinib for 9 days were lysed and the indicated proteins were detected by western blotting with immunoprecipitation of the indicated proteins. **d** PC-9, HCC4011 parental cells, and DT cells generated by treatment with 3 μmol/L osimertinib were treated with 1 μmol/L of NPS1034, OSI906, or a combination of these agents for 72 h and the cell viability was assessed using MTT assays. **e** The indicated cells were incubated with osimertinib (3 μmol/L) with or without NPS1034 (1 μmol/L) for 1 h. The cells were lysed and the indicated proteins were detected by western blotting. **f** Cells were treated with DMSO, 100 nmol/L osimertinib, 1 μmol/L NPS1034, or a combination of 100 nmol/L osimertinib and 1 μmol/L NPS1034 for 3 weeks with the drugs replenished every 72 h. The plates were stained with crystal violet and visually examined. A plate representative of three independent experiments is shown

combined with osimertinib treatment in a cell-line-derived xenograft (CDX) model using high-AXL-expressing PC-9 cells. Using this approach, we evaluated the effect of NSP1034 using two different treatment schedules, (1) during the initial phase and (2) during the tolerant phase. For the initial phase studies, mice were continuously administered osimertinib alone, NSP1034 alone, or a combination of osimertinib and NSP1034 for 5 days a week by oral gavage until the end of experiment (Fig. 7a). Treatment with NSP1034 alone had no effect on the growth of PC-9 tumors. Treatment with osimertinib alone caused tumor regression within 1 week, but the tumors re-grew within 7 weeks, indicating rapid recurrence due to acquired resistance. Combined treatment with osimertinib and NSP1034 also caused tumor regression within 1 week and the size of the regressed tumors was maintained for 10 weeks. These results indicated that combined

treatment with osimertinib and NSP1034 commenced during the initial phase prevented the rapid growth of high-AXL-expressing *EGFR*-mutated NSCLC cells in vivo. No apparent adverse events, including weight loss, were observed during these treatments (Supplementary Figure 16B). For the tolerant phase studies, all the PC-9 tumor-bearing mice were initially treated with osimertinib alone and the tumor size regressed within 1 week (Fig. 7b). On day 8, the mice were randomly divided into two groups. One group was treated with osimertinib alone and the other was treated with both osimertinib and NSP1034. The tumors treated with osimertinib alone re-grew, again showing rapid recurrence. The re-growth of the tumors treated with osimertinib and NSP1034 was much slower than those treated with osimertinib alone. Overall, the results from the rapid recurrence model using PC-9 cells indicated that the combined treatment of

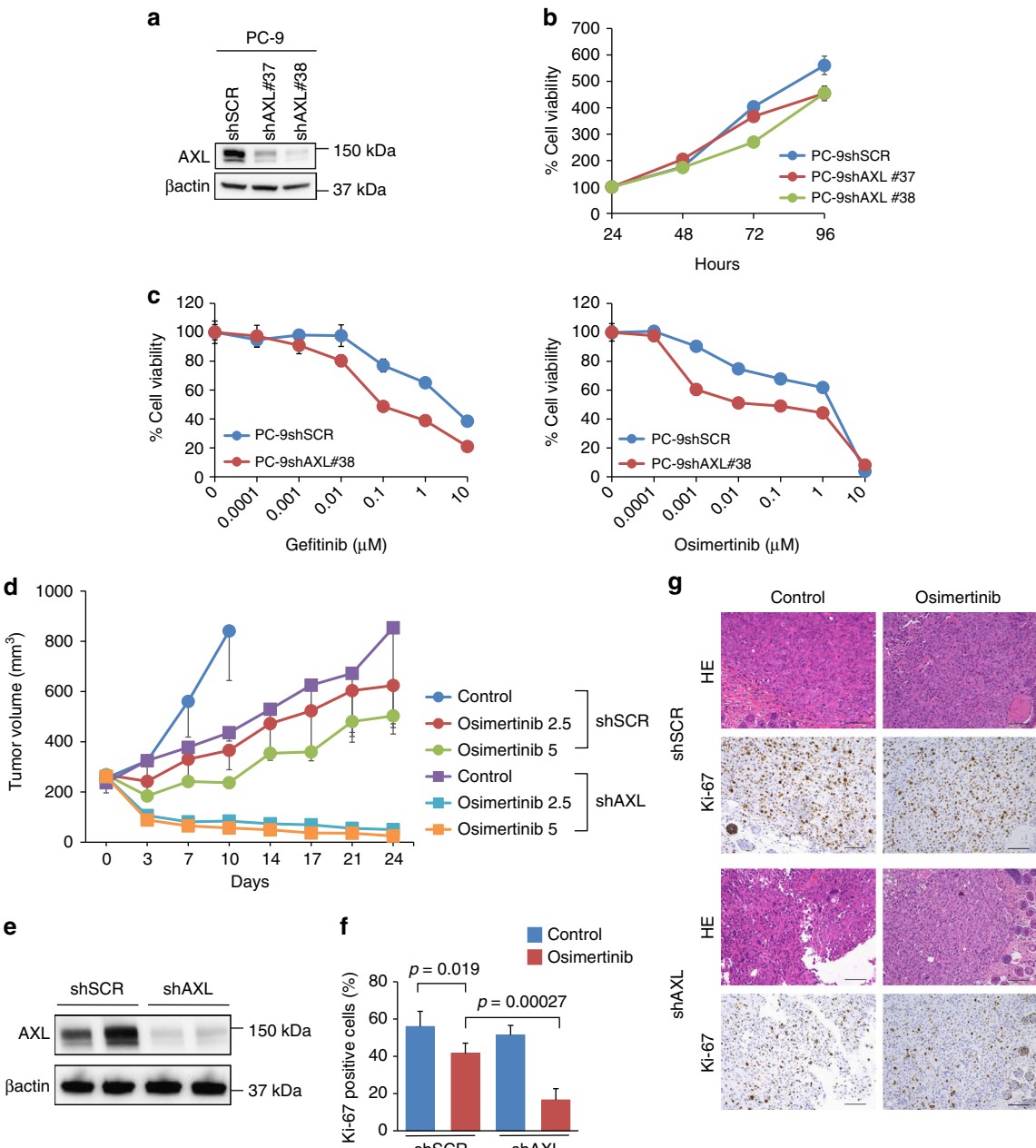

**Fig. 6** AXL inhibition shrank PC-9 tumors treated with osimertinib in vivo. Stable PC-9 cell lines were generated by the introduction of short hairpin RNAs (shRNA) that mediated inhibition of AXL expression (#37 and #38) and control nontargeting (SCR) shRNA. **a** The cells were lysed and the indicated proteins were detected by western blotting. **b** Cells were incubated for the indicated times and their viability assessed using MTT assays. **c** Cells were incubated with gefitinib or osimertinib at the indicated concentrations for 72 h and the cell viability assessed using MTT assays. **d** Following the subcutaneous injection of the indicated cells into nude mice, vehicle (control) or osimertinib (2.5 or 5 mg/kg) were administered. Tumor volumes were determined and the results are plotted over time from the start of treatment (mean ± SEM). **e** Western blotting analysis of the presence of the indicated proteins from the harvested tumors as described for (**d**). **f** Quantification of proliferating cells, as determined by their Ki-67-positive proliferation index (percentage of Ki-67-positive cells) as described in the Methods. Columns, mean of five evaluated areas; bars, SD. Comparisons by paired Student's *t* tests. **g** Representative images of PC-9 xenografts containing the indicated shRNAs following immunohistochemical staining with antibodies specific for human Ki-67. Bar, 100 μm

osimertinib with NSP1034 commenced during either the initial phase or tolerant phase prevented tumor re-growth following osimertinib treatment.

**AXL inhibitor combined with osimertinib stabilized PDX tumor**. Finally, using two different treatment schedules we evaluated the effect of NSP1034 against patient-derived xenograft tumors that highly expressed the AXL protein (Fig. 7c). In

treatment during the initial phase, NSP1034 alone had no effect on the growth of PDX tumors (Fig. 7d). Treatment during the initial phase with osimertinib alone slightly delayed tumor growth compared to tumors in control animals, but the tumors never regressed, indicating that the PDX tumors were intrinsically resistant to osimertinib. Under these experimental conditions, combined treatment with osimertinib and NSP1034 resulted in the stabilization of tumor growth. No apparent adverse events,

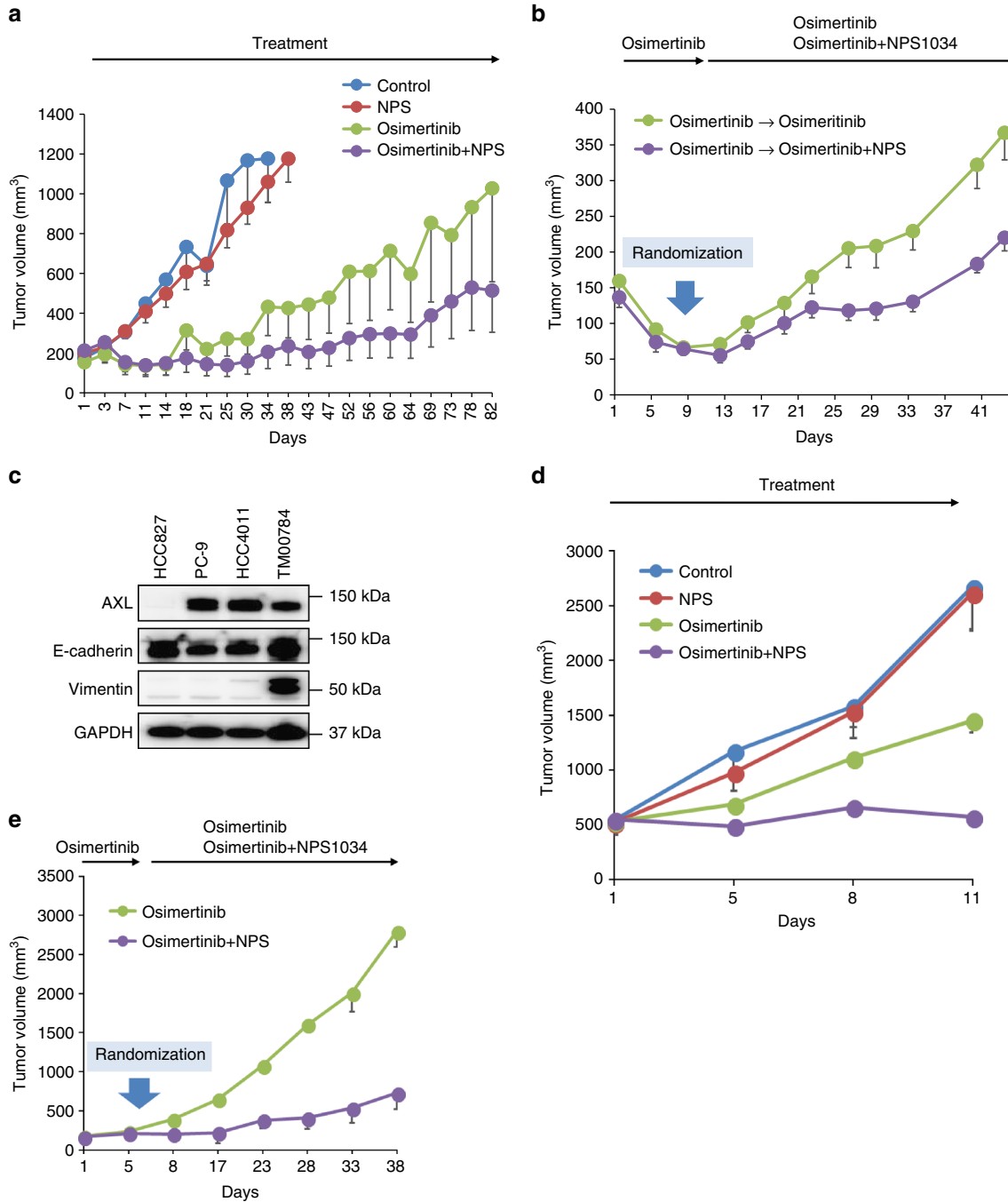

**Fig. 7** AXL inhibitor with osimertinib inhibited growth of AXL-high tumors in vivo. **a** PC-9 cell-line-derived xenograft (CDX) tumors were treated with vehicle (control), NPS1034 50 mg/kg, osimertinib 5 mg/kg, or NPS1034 50 mg/kg plus osimertinib 5 mg/kg (*n* = 6 each). Tumor volumes were measured over time from the start of treatment and the results are shown (mean ± SEM). **b** PC-9 CDX tumors were treated with osimertinib (5 mg/kg) for 8 days followed by the continuous administration of 5 mg/kg osimertinib or in combination with 50 mg/kg NPS1034 (*n* = 7 each) administered. The results of tumor volume are plotted (mean ± SEM). **c** The indicated cells and patient-derived xenograft (PDX) tumors (TM00784) were lysed and the indicated proteins were detected by western blotting. **d** PDX tumors (TM00784) were untreated (control), treated with 50 mg/kg NPS1034, 5 mg/kg osimertinib, or a combination of 50 mg/kg NPS1034 and 5 mg/kg osimertinib (*n* = 4 each). The results of tumor volume are plotted (mean ± SEM). **e** PDX tumors (TM00784) were treated with 5 mg/kg osimertinib for 7 days followed by continuous administration of 5 mg/kg osimertinib or in combination with 50 mg/kg NPS1034 (*n* = 4 each). The results of tumor volume are plotted (mean ± SE)

including weight loss, were observed during these treatments (Supplementary Figure 16C). In treatment during the tolerant phase, all PDX tumor-bearing mice were initially treated with only osimertinib and the tumor sizes increased within 1 week (Fig. 7e). On day 8, the mice were randomly divided into two groups. The group of mice with PDX tumors treated continuously

with osimertinib alone showed further tumor growth at a rate consistent with its intrinsic resistance to osimertinib. On the other hand, the combined treatment with both osimertinib and NSP1034 stabilized the sizes of the PDX tumors and remarkably prevented further enlargement. These results indicated that combined treatment with NSP1034 started either during the

initial phase or tolerant phase demonstrated efficacy at stabilizing PDX tumors in the intrinsic resistance model.

## Discussion

The efficacy of TKI treatment of tumors with driver oncogenes is limited due to the reactivation of specific signaling pathways via multiple feedback mechanisms, demonstrating the need to develop therapeutic approaches to effectively treat these cancers[28,29]. The current study was performed to investigate the major feedback mechanisms that emerged as a result of exposure of NSCLC to osimertinib, identifying the occurrence of an adaptive response due to the activation of AXL, which binds to the HER family proteins EGFR and HER3 through a negative feedback loop involving the suppression of SPRY4. We previously demonstrated the role of SPRY4 in mesenchymal-like *KRAS*-mutant lung cancers in which FGFR1 is dominantly expressed but suppressed by the negative regulatory sprouty proteins, including SPRY4 [28]. This is the first report that SPRY4 might be a potential negative regulator of AXL signaling in lung cancer with driver oncogenes that is treated with osimertinib.

AXL has been reported to promote EGFR-induced signaling by binding to EGFR and other members of the HER family including MET and PDGFR, resulting in acquired resistance to EGFR-TKIs[30]. In addition, the activation of AXL signaling has been shown to have varying cell-specific and tissue-specific effects during treatment with EGFR-targeted agents. For instance, in wild-type *EGFR* cancer cells, AXL overexpression induces resistance to the anti-EGFR antibody cetuximab via MAPK signaling[31]. In contrast, the primary escape mechanism from osimertinib treatment in the *EGFR*-mutated NSCLC cells was through a downregulation of SPRY4, which predominantly activates the AXL-AKT axis. The precise mechanism remains unclear because the specific scaffold protein related to AXL activation was not detected. We plan to conduct further experiments to identify these proteins and further define the mechanism.

EMT with high-AXL expression is reported to be associated with acquired resistance to various drugs, including EGFR-TKIs[21]. It is unclear, however, whether AXL is an inducer or effector of the EMT process and whether AXL inhibition, combined with an EGFR-TKI, is able to overcome the acquired resistance to EGFR-TKIs induced by EMT, a feature that may be closely associated with tumor heterogeneity[21,32,33]. AXL inhibition is shown to be pivotal in overcoming intrinsic or acquired resistance to BRAF/MEK inhibitors by melanoma harboring *NRAS* and *BRAF* mutations[34]. Therefore, the present study focused on the role of AXL during the initial treatment of *EGFR*-mutated NSCLC cells in an effort to prevent the development of resistant clones via AXL signaling, which may be involved in the promotion of EMT. Cell-based assays showed that osimertinib treatment in combination with AXL inhibitors initiated during the initial phase had greater inhibitory effects on *EGFR*-mutated NSCLC cells. Moreover, in vivo experiments demonstrated that this combination prevented the re-growth of CDX tumors in a rapid recurrence model for osimertinib and stabilized PDX tumors in an intrinsic resistance model for osimertinib. To our knowledge, this report is the first to demonstrate the efficacy and tolerability of osimertinib combined with an AXL inhibitor in the treatment of *EGFR*-mutated NSCLC. The results also suggested that this approach might prevent escape by acquired resistance to osimertinib by these cancers.

Despite showing a good initial response to targeted molecular therapy, a small percentage of cells survived and expanded, resulting in acquired resistance to EGFR-TKIs. These drug-tolerant cells result in tumor heterogeneity, enhancing tumor

recurrence[35–37]. A recent study showed that GPX4 inhibition induced ferroptosis, an iron-dependent form of cell death, in various types of persistent cancer cells, which acquire a dependency on GPX4 and resistance to specific targeted molecular therapy[38]. While a small subpopulation of DT cells is reported to maintain viability via IGF-1R signaling in *EGFR*-mutated NSCLC treated with first EGFR-TKIs[14], the current study also demonstrated AXL binding to EGFR and IGF-1R in DT cells treated with osimertinib, and thus AXL activation was greater in DT cells than in the parental cells. Treatment with an AXL or IGF-1R inhibitor alone reduced the viability of cells that were tolerant to osimertinib, suggesting that the DT cells may switch to the AXL-IGF-1R axis, independent of the AXL-HER family axis. Importantly, we demonstrated here that the exposure of *EGFR*-mutated NSCLC cells to osimertinib caused a dynamic change of specific related proteins via their signal transduction in order for the cells to survive and escape from cell death. Unfortunately, we were unable to identify the specific EGFR proteins that bound to AXL and that were crucial for drug tolerance. These observations clearly demonstrated that AXL signaling played crucial roles in *EGFR*-mutated NSCLC cells regarding both the initial adaptive response to osimertinib and in the development of tolerance.

Aside from the *EGFR*-T790M-resistant mutations, promising biomarkers for acquired resistance to first-generation EGFR-TKIs have not been identified. Therefore, it may not be worthwhile to evaluate all patients with acquired resistance to EGFR-TKIs to determine which have the potential for AXL inhibition. We found that AXL overexpression was associated with a poor response to osimertinib, as well as to the first- and second-generation EGFR-TKIs in the clinical setting, suggesting that overexpression of AXL may be a novel biomarker for the initial tolerance of *EGFR*-mutated NSCLCs to EGFR-TKIs. The limitation of our study is the small number of clinical specimens used in Fig. 2d. Larger scale studies are, therefore, warranted to further clarify the correlation between AXL expression and the clinical response to osimertinib in EGFR-TKI naïve *EGFR*-mutated NSCLCs in the future.

In conclusion, this study showed that high levels of AXL activation had biological significance in EGFR-driven NSCLC cells treated with the EGFR-TKI osimertinib. Our findings demonstrated a pivotal role for AXL in the intrinsic resistance of *EGFR*-mutated lung cancer to osimertinib and the emergence of osimertinib-tolerant cells. These results suggest that treatment during the initial phase with a combination of osimertinib and an AXL inhibitor may prevent the development of intrinsic resistance to osimertinib and the emergence of drug-tolerant cells in *EGFR*-mutated lung cancer overexpressing AXL.

## Methods

**Cell cultures and reagents**. Nine human NSCLC cell lines with mutations in *EGFR* were utilized. The human NSCLC cell lines HCC4011 and H3255 were generously provided by Drs. David P. Carbone (Ohio State University Comprehensive Cancer Center, Columbus, OH) and John D. Minna (University of Texas Southwestern Medical Center, Dallas, TX), respectively. The H1975 human lung adenocarcinoma cell line with the *EGFR*-L858R/T790M double mutation was kindly provided by Drs. Yoshitaka Sekido (Aichi Cancer Center Research Institute, Japan) and John D. Minna. The human cell lines HCC827, HCC4006, and H1650 were purchased from the American Type Culture Collection (Manassas, VA), and the PC-9 cell line was obtained from RIKEN Cell Bank (Ibaraki, Japan). The PC-9KGR cells, which contain deletions in the *EGFR* exon 19 and the T790M mutation, were developed from PC-9 cells by stepwise exposure to gefitinib[39]. The PC-9GXR cells, which contain deletions in the *EGFR* exon 19 and the T790M mutation, were established at Kanazawa University (Kanazawa, Japan) from PC-9 cell xenograft tumors in nude mice that had acquired resistance to gefitinib. All of these cell lines were maintained in Roswell Park Memorial Institute (RPMI) 1640 medium (GIBCO, Carlsbad, CA) with 10% fetal bovine serum (FBS), penicillin (100 U/mL), and streptomycin (50 g/mL) in a humidified $CO_2$ incubator at 37 °C. All cells were passaged for less than 3 months before being renewed with frozen,

early-passage stocks. Cells were regularly screened for mycoplasma using a MycoAlert Mycoplasma Detection Kit (Lonza). Cell lines were authenticated by DNA fingerprinting. Gefitinib, osimertinib, rociletinib, NPS1034, OSI-906, buparlisib, trametinib, and caffeic acid phenethyl ester (CAPE) were obtained from Selleckchem (Houston, TX), and ASP2215 was kindly provided by Astellas Pharma Inc. (Tokyo, Japan).

**Human phospho-kinase antibody array.** The relative levels of phosphorylation of 43 kinases and two related total proteins were measured using the Human Phospho-Kinase Array Kit (R&D Systems), using a modification of the manufacturer's instructions. Briefly, cells were cultured in RPMI-1640 containing 10% FBS and lysed in array buffer prior to reaching confluence. The arrays were blocked with blocking buffer and incubated with 450 µg of cell lysate overnight at 4 °C. The arrays were washed, incubated with a horseradish peroxidase (HRP)-conjugated phospho-kinase antibody, and treated with SuperSignal West Dura Extended Duration Substrate Enhanced Chemiluminescent Substrate (Pierce Biotechnology, Rockford, IL). Each experiment was independently performed at least twice.

**Antibodies and western blotting.** Protein aliquots of 25 µg each were resolved by SDS polyacrylamide gel electrophoresis (Bio-Rad, Hercules, CA) or 1000 µg aliquots of total proteins were immunoprecipitated with the appropriate antibodies. The immune complexes were recovered with Protein G-Sepharose or Protein A-Sepharose beads (Zymed Laboratories, California). Electrophoresed protein samples or immunoprecipitated samples were transferred to polyvinylidene difluoride membranes (Bio-Rad). After washing three times, the membranes were incubated with blotting-grade blocker (Bio-Rad) for 1 h at room temperature and overnight at 4 °C with primary antibodies to p-AXL (Tyr702), t-AXL, p-EGFR, p-MET, t-MET, p-HER3 (Tyr1289), t-HER3, p-IGF-1R, t-IGF-1R, p-Akt (Ser473), t-Akt, E-cadherin, N-cadherin, Vimentin, ALDH1A1, CD44, β-actin (13E5) (1:1000 dilution; Cell Signaling Technology, Danvers, MA, USA), p-Erk1/2 (Thr202/Tyr204), t-Erk1/2, t-EGFR (1:1000 dilution, R&D Systems), SPRY4 (1:1000 dilution; Proteintech, Rosemont, IL).

After washing three times, the membranes were incubated for 1 h at room temperature with HRP-conjugated species-specific secondary antibody. Immunoreactive bands were visualized using SuperSignal West Dura Extended Duration Substrate Enhanced Chemiluminescent Substrate (Pierce Biotechnology). All of the uncropped western blots with molecular weight indicated were presented in Supplementary Figure 17. Each experiment was independently performed at least three times.

**Cell viability assay.** Cell viability was determined using the MTT (3-(4,5-Dimethylthial-2-yl)-2,5-Diphenyltetrazalium Bromide) dye reduction method. Briefly, tumor cells ($2–3\times10^3$ cells/100 µL/well) in RPMI 1640 medium supplemented with 10% FBS were plated in 96-well plates and cultured with the indicated compound for 72 h. After culturing, 50 µg of MTT solution (2 mg/mL, Sigma, St. Louis, MO) was added to each well. Plates were incubated for 2 h, the medium was removed, and the dark blue crystals in each well were dissolved in 100 µL of dimethyl sulfoxide (DMSO). Absorbance was measured with a microplate reader at a test wavelength of 550 nm and a reference wavelength of 630 nm. The percentage of growth was determined relative to untreated controls. Experiments were repeated at least three times with triplicate samples.

**Transfection of siRNAs.** Duplexed Silencer® Select siRNAs for *AXL* (s1845 and s1846), *EGFR* (s565), and *SPRY4* (s37826), and Stealth RNAi for *MET* (HSS106478), *SPRY4* (HSS130078), and *HER3* (HSS140802) were purchased from Invitrogen (Carlsbad, CA) and transfected into cells using Lipofectamine RNAi-MAX (Invitrogen) in accordance with the manufacturer's instructions. In all experiments, Silencer® Select siRNA for Negative Control no.1 (Invitrogen) was used as the scrambled control. Knockdown of *AXL*, *EGFR*, *MET*, *HER3*, and *SPRY4* were each confirmed by western blotting analysis. Each sample was tested in triplicate with three independent experiments being performed.

**Plasmid construction.** $2\times10^5$ PC-9 cells were seeded in six-well plates. The next day, the cells were transfected with 2.5 µg DNA of pIRESpuro2 AXL (Addgene) using 5 µL Lipofectamine LTX Reagent (Invitrogen) and 2.5 µL PLUS Reagent (Invitrogen) in 250 µL serum-free Opti-MEM (Life Technologies). After 4 h, culture medium were exchanged and 24 h later, puromycin (Sigma-Aldrich) was added to the medium at the final concentration of 1 µg/mL.

**Lentivirus and infections.** In knockdown experiments, shRNAs in lentivirus expression vectors included MISSION pLKO.1 constructs (Sigma-Aldrich) specific for AXL (clones NM_021913.TRCN0000001037 and NM_021913. TRCN0000001038), SPRY4 Precision LentiORF (PLOHS_ccsbBEn_04257), and a scrambled control (SHC001).

**Cell-line-derived xenograft (CDX) models.** Suspensions of $5\times10^6$ cells were injected subcutaneously into the flanks of 5-week-old male mice with severe combined immunodeficiency obtained from Clea Japan (Tokyo, Japan). Once the

mean tumor volume reached approximately 100–300 mm³, five mice each were injected with PC-9-shSCR or PC-9-shAXL, six with PC-9 cell-line-derived xenografts (CDX) at the time of initial treatment, and seven with PC-9 CDX at the time of sequential therapy. Drugs were administered 5 or 7 days a week by oral gavage and their body weight and general condition were monitored daily. Tumors were measured twice weekly using calipers and their volumes were calculated as width$^2$ × length/2. The study protocol was approved by the Ethics Committee on the Use of Laboratory Animals and the Advanced Science Research Center, Kanazawa University, Kanazawa, Japan (approval no. AP-122505). According to institutional guidelines, surgery was performed after the animals were anesthetized with sodium pentobarbital and efforts were made to minimize animal suffering

**Patient-derived xenograft (PDX) models.** Xenografts from a 42-year-old woman with lung cancer containing the *EGFR* L858R mutation (TM00784) were purchased from The Jackson Laboratory (Bar Harbor, ME). Female NOD.Cg-Prkdcscid Il2rgtm1Wjl/SzJ (NSG) mice aged 6–8 weeks were engrafted with tumor fragments at passage P7 at the Jackson Laboratory. The mice were transferred to the animal facility at Kanazawa University and randomized once their mean tumor volume reached approximately 500 mm³ or 150–200 mm³. The study protocol was approved by the Ethics Committee on the Use of Laboratory Animals and the Advanced Science Research Center, Kanazawa University, Kanazawa, Japan.

**Cell-cycle assay.** The cell-cycle was determined using a FACSCalibur flow cytometer (BD Biosciences, San Diego, CA) with a BD Pharmingen™ BrdU Flow Kit, which allows for the detection and quantification of the percentage of cells in each phase of cell cycle (G1, S, and G2-M) using labeled bromodeoxyuridine (BrdU) and 7-amino-actinomycin (7-AAD) staining. Each experiment independently repeated at least three times.

**DNA extraction.** DNA was extracted from frozen tissue samples according to the manufacturer's recommendations with slight modifications. Briefly, each sample was placed in a sterile tube with lysis buffer (Promega, Madison, WI) and mechanically disrupted using a TissueLyser TL system (Qiagen, Venlo, Netherlands) for 1 min at 50 Hz. Proteinase K (Promega) and RNaseA (Sigma-Aldrich Corporation, St. Louis, MO) were added to each sample and mixed by vortexing (Scientific Industries Inc, Bohemia, NY, USA). The samples were then incubated for 1 h at 56 °C. DNA was isolated from frozen tissue samples and cell lines using Maxwell®16 Blood DNA Purification Kits and a Maxwell 16 Instrument (Promega) according to the manufacturer's instructions. DNA was eluted in 50 µL of nuclease-free water and the concentration measured using Quant-iT PicoGreen® dsDNA Assay kits (Thermo Fisher Scientific, Waltham, MA).

**Patients.** Specimens of tumors containing *EGFR*-activating mutations prior to the initial treatments with EGFR-TKIs gefitinib, erlotinib, afatinib, or ASP8273 were obtained from 46 lung adenocarcinoma patients hospitalized at Kanazawa University Hospital (Kanazawa, Japan), Niigata University Hospital (Niigata, Japan), Niigata Cancer Center Hospital (Niigata, Japan), Nagasaki University Hospital (Nagasaki, Japan), or the Japanese Red Cross Nagasaki Genbaku Hospital (Nagasaki, Japan). Specimens of tumors containing *EGFR*-activating mutations, prior to treatment with osimertinib, were obtained from 11 lung adenocarcinoma patients hospitalized at the Kanazawa University Hospital (Kanazawa, Japan), Japanese Red Cross Kyoto Daiichi Hospital (Kyoto, Japan), Niigata University Hospital (Niigata, Japan), and Niigata Cancer Center Hospital (Niigata, Japan). All patients were participants in the Institutional Review Board of Kanazawa University, Niigata University, Niigata Cancer Center, Nagasaki University, the Japanese Red Cross Nagasaki Genbaku Hospital, and Japanese Red Cross Kyoto Daiichi Hospital-approved studies and all provided written informed consent.

**Histological analyses of tumors.** Formalin-fixed, paraffin-embedded tissue sections (4 µm thick) were deparaffinized. Antigen was retrieved by microwaving the tissue sections in 10 mM citrate buffer (pH 6.0). Proliferating cells were detected by incubating the tissue sections with Ki-67 antibody (Clone MIB-1; DAKO Corp, Glostrup, Denmark). Based on the expression patterns, tumor cells in tissue specimens were separately evaluated for expression of AXL using an anti-AXL antibody (1:200; goat polyclonal, R&D Systems). Because immunohistochemical studies have shown that AXL is present primarily in the cytoplasm of cells and that its staining varies in intensity, we quantified its expression as negative (0), weak (1+), moderate (2+), and strong (3+) compared with vascular endothelial cells as an internal control[18]. After incubation of the specimens with the secondary antibody and treatment with using the Vectastain ABC Kit (Vector Laboratories, Burlingame, CA), peroxidase activity was visualized using 3,3′-diaminobenzidine as a chromogen. The sections were counterstained with hematoxylin.

**Quantification of immunohistochemistry results.** The five areas containing the highest numbers of positively stained cells within each section were selected for histologic quantitation using light or fluorescent microscopy at a 400-fold magnification.

**Statistical analysis**. Data from the MTT assays and tumor progression of xenografts were expressed as means ± standard deviation (SD) and as means ± standard error (SE), respectively. The statistical significance of differences was analyzed using one-way ANOVA and Spearman rank correlations. PFS and 95% confidence intervals were determined using the Kaplan−Meier method and compared using the log-rank test. HRs of clinical variables for PFS were determined using a univariate Cox proportional hazards model. All statistical analyses were performed using GraphPad Prism Ver. 6.0 (GraphPad Software, Inc., San Diego, CA, USA), with a two-sided $P$ value less than 0.05 being considered statistically significant.

**Reporting Summary**. Further information on experimental design is available in the Nature Research Reporting Summary linked to this Article.

## Data availability

The data that support the findings of this study are available from the authors upon reasonable request.

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

## Acknowledgements

We appreciate the generosity of Dr. David P. Carbone (The Ohio State University Comprehensive Cancer Center, Columbus, OH) for providing the HCC4011 cells, Dr. John D. Minna (University of Texas Southwestern Medical Center) for providing the H3255 cells, Drs. Yoshitaka Sekido (Aichi Cancer Center Research Institute, Japan) and John D. Minna (University of Texas Southwestern Medical Center) for providing the H1975 cells, and Astellas Pharma Inc. (Tokyo, Japan) for kindly providing the ASP2215. This study was supported by a research grant for developing innovative cancer chemotherapy from the Kobayashi Foundation for Cancer Research (to T.Y.), a Grant for Lung Cancer Research funded by the Japan Lung Cancer Society (to T.Y.), grants from the Japan Society for the Promotion of Science (JSPS) KAKENHI grant number 16K19447, 16H05308 (to S.Y.), the Project for Cancer Research and Therapeutic Evolution (P-CREATE) grant number 16cm0106513h0001 (to S.Y.), and Extramural Collaborative Research Grant of Cancer Research Institute, Kanazawa University (to T.Y. and to H. Taniguchi).

## Author contributions

T.Y. and S.Y. supervised the study. H. Taniguchi, T.Y., S.T., and S.Y. conceived and designed the experiments. H. Taniguchi, T.Y., R.W., K.Tanimura., Y.A., and A.T. performed the experiments. T.Y., L.H.A., M.B., and H.U. performed data analysis. T.Y.,

A.N., A.Y., S.S., I.M., S.W., T.Kitazaki., S.M., H. Tanaka, T.Kikuchi., H.Y., H.M., J.U., K.Takayama., and S.Y. contributed reagents and materials. H. Taniguchi, T.Y., and S.Y. wrote the manuscript. All authors reviewed and approved the manuscript.

## Additional information

**Competing interests:** S.Y. obtained commercial research grants from AstraZeneca, Chugai Pharm, and Boehringer-Ingelheim, and has received speaking honoraria from AstraZeneca, Chugai Pharma, and Boehringer-Ingelheim. The remaining authors declare no competing interests.

