## [Peer Review File · Nature Communications]

Reviewers' comments:

Reviewer #1 (Remarks to the Author):

This study reports on the role of AXL activation in limiting response to EGFR inhibitor treatment (osimertinib) in lung cancer. The findings are interesting, novel, and point towards the clinical development of AXL inhibitors in combination with osimertinib in lung cancer patients. There are issues to address before publication:

(1) The authors should validate the target of the AXL inhibitor (NPS) by making a drug-resistance form of AXL that is not inhibited by NPS treatment and showing that this blocks the effects of the drug.

(2) The authors need to increase the number of clinical specimens showing that high AXL is correlated with worse response to osimertinib.

(3) The authors need to clarify which pathway and phenotypes that AXL controls to exert osimertinib resistance (for instance, ERK, AKT, NFkappaB, EMT, stem cell-like), and examine the clinical specimen for the associated relevant molecular phenotypes in association with osimertinib resistance.

(4) Is the role of AXL specific for osimertinib, or does this relate to other 3rd generation EGFR TKIs as well?

(5) Is the AXL-mediated resistance reversible?

Reviewer #2 (Remarks to the Author):

As indicated in the title, the authors propose here that AXL confers intrinsic resistance to osimertinib, a new generation EGFR-TKI, and promote the emergence of tolerant lung cancer cell. Since this drug will be used as the first-line drug for EGFR-muted lung cancer, one of the most frequent cancer in never smokers, the topic of this study is highly significant to consider the future of precision cancer medicine using highly specific and efficient drugs. The contents are very

important and attractive, however, the reviewer feels concerns on the following points. To clarify these points is necessary to draw a conclusion and also to publish this manuscript.

1. Results: “However, approximately 20–30% of the PC-9 cells survived, even when treated with a high concentration (10 microM) of osimertinib for 72 h (Supplemental Figure 1). To determine the mechanism by which these cells escaped the effects of osimertinib,..”. This is the start point of this study to consider “intrinsic resistance” in a cancer cell population. However, this finding has not been reproduced throughout this study, as presented in Figure 4A, 5A, 6C, and Supplemental Figure 4. Is it really a basis of this study that some PC9 cells escape the effects of osimertinib under this condition?

2. In abstract, the authors say “Activated AXL was associated with EGFR and HER3 in maintaining cell survival and inducing the emergence of cells tolerant to osimertinib”. However, association of AXL with HER3 were not examined in the experiment to see the emergence of DT cells (Figure 5B). In addition, effect of drug-mediated EGFR and AXL co-inhibition on HER3 phosphorylation was examined either. Therefore, the results are not conclusive.

3. The experiments using a panel of nine EGFR-mutated lung cancer cell lines in Figures 2 and 3 are interesting and important to generalize the finding of PC9 cell. However, the criteria to divide them into AXL-high and –low cell lines is ambiguous. The authors describe “EGFR phosphorylation tended to be higher in cells with the lower relative AXL expression levels,.. (page 10)”. The reviewer thinks amounts of EGFR protein rather than phosphorylation show a correlation with the AXL subgroup. How about the expression level of SPRY4? Effect of EGFR and AXL co-inhibition on AKT phosphorylation was not examined in AXL-low cells in Figure 3, so, it is unclear whether this phenomenon is responsible for the differential drug response in Figure 3B or not.

4. In Table 2C, unfortunately, it is unclear whether osimertinib show the same tendency or not, although most of the experiments in this study have been done with osimertinib.

5. Involvement of AXL in acquired TKI resistance has been suggested by several studies as instructed in Introduction. This fact reduces the novelty/significance of this study focusing on intrinsic TKI resistance, since it is conceivable that both mechanisms overlap with each other, as indicated by T790M mutation.

6. “While AXL was not constitutively phosphorylated, its phosphorylation was induced by osimertinib at 4 h and increased through 72 h (Page 7).... These results suggested that osimertinib exposure may have activated AXL and thereby re-activated HER3, MET, and EGFR in PC-9 cells (Page 8).” These

results indicate that AXL is a key trigger for resistance emergence. However, in Supplemental Figure 2A, phosphorylation of AXL is not likely to be induced at 4hr after osimertinib treatment. The reviewer thinks that data in Supplemental Figure 2A is important to explain the resistance mechanism through AXL.

Minor points:

1. Abstract: "TKI" must be spelled out.
2. The authors should explain the specificity of osimertinib against other kinases than mutated EGFR. How about its effect of the inhibitor on other kinases such as AXL, AKT1 and HER3? Since a variety fields of researchers read Nat Com papers, such a basic explanation is needed.
3. Page 9: The authors use the term "adversely activated". The reviewer does not understand why and how the activation is adverse.

Reviewer #1 (Remarks to the Author):

(1) The authors should validate the target of the AXL inhibitor (NPS) by making a drug-resistance form of AXL that is not inhibited by NPS treatment and showing that this blocks the effects of the drug.

Reply:

We agree that the validation of the target of NPS1034 is scientifically very important. However, to the best of our knowledge, no resistance mutation in AXL to NPS1034 has been reported. In our study, NPS1034 sensitized the AXL high-expressing, *EGFR*-mutated NSCLC cells to osimertinib and prevented the production of drug tolerant (DT) cells. To obtain the AXL-resistant form, we needed to culture the *EGFR*-mutated NSCLC cell lines in the presence of both osimertinib and NPS1034. Theoretically, under these culture conditions, the cells could become resistant, even if they only acquired osimertinib-resistance mutations, such as *EGFR*-C797S, without an AXL mutation. Therefore, the production of a drug-resistant form of AXL was really challenging within 3 months (the deadline for the revision).

Alternatively, to prove that AXL was the target of NPS1034 for the sensitization of *EGFR*-mutated NSCLC cells to osimertinib, we examined the effect of wild-type AXL overexpression in PC-9 cells treated with osimertinib and NPS1034. While wild-type AXL overexpression did not affect the viability of PC-9 cells treated with NPS1034 alone, it remarkably induced osimertinib resistance in the presence and absence of NPS1034. These new findings indicate that the target of NPS1034 was AXL in our study. These data are as only for reviewer Figure 1.

(2) The authors need to increase the number of clinical specimens showing that high AXL is correlated with worse response to osimertinib.

Reply:

As recommended by the reviewer, we collected new clinical specimens from the *EGFR*-mutated NSCLC patients treated with osimertinib. We could obtain IRB approval for these additional experiments at the four institutes by October 2018. Osimertinib was first approved for T790M-positive *EGFR*-mutated NSCLC patients previously treated with *EGFR*-TKIs in May of 2016, and in August of 2018, it was approved as the first line treatment for patients with untreated *EGFR*-mutated NSCLC in Japan. We could collect 11 tumor specimens from *EGFR*-mutated NSCLC patients at these four institutes and examined the correlation between AXL expression and the outcome after osimertinib treatment in these patients. All specimens were obtained before the initiation of *EGFR*-TKI treatment, and all were positive for the *EGFR*-T790M mutation at the acquisition of resistance to the initial *EGFR*-TKI treatment. No tumor specimens could be collected from the patients who were treated with

osimertinib as the first line of treatment, because the clinical response has not been determined at this time point.

In this new set of specimens, high (3+), intermediate (2+), low (1+), and no (0) AXL expression was observed in three (27.2%), one (18.2%), seven (63.6%), and zero (0.0%) specimens, respectively. The response rate for the patients with AXL expression scores of 1+ to 2+ was high (85.7% to 100%), whereas, for the patients with AXL expression scores of 3+, the response rate was relatively lower (66.7%) (Figure 2D). Moreover, the PFS for osimertinib treatment trended toward being shorter in the patients whose AXL expression scores were 3+, compared to those whose AXL expression scores were 0 to 2+ ($P = 0.449$ and HR = 0.81) (Supplemental Figure 6C).

Although the number of patients with an AXL score of 3+ was very limited, these results were consistent with those in shown Figure 2C. Larger scale studies are warranted to further clarify the correlation between AXL expression and the clinical response to osimertinib in EGFR-TKI naïve *EGFR*-mutated NSCLC in the future. We have added these statements in the Results (page 11, lines 10-13, 16-17, 18; page 12, lines 1-12), Discussion (page 24, lines 9, 11-13), and Materials and methods sections (page 31, lines 13-17).

(3) The authors need to clarify which pathway and phenotypes that AXL controls to exert osimertinib resistance (for instance, ERK, AKT, NFkappaB, EMT, stem cell-like), and examine the clinical specimen for the associated relevant molecular phenotypes in association with osimertinib resistance.

Reply:

As recommended by the reviewer, we investigated the expression of stem cell markers (ALDH1A1 and CD44) and EMT markers (E-cadherin and vimentin) in *EGFR*-mutated NSCLC cells (PC-9 and PC-9GXR) treated with or without osimertinib and/or NPS1034 for 72 h. The expression of these molecules was not remarkably affected by osimertinib with or without NPS1034 (only for reviewer Figure 2).

We have shown that ERK protein expression is not affected by 72 h of exposure to osimertinib and that ERK phosphorylation is inhibited by osimertinib up until 72 h in PC-9 cells (Figure 1B). We further examined the role of ERK-mediated signaling, and evaluated the effect of an MEK inhibitor (trametinib). While trametinib alone slightly inhibited the viability of PC-9 cells, the inhibitory effect of trametinib combined with osimertinib was weaker than that of NPS1034 combined with osimertinib (Supplemental Figure 10). These results indicate that the inhibitory effect of NPS1034 combined with osimertinib is not predominantly mediated by ERK.

We also evaluated the effect of an NF- κ B inhibitor (CAPE). CAPE did not discernibly affect the viability of PC-9 cells, irrespective of the presence of osimertinib (Supplemental Figure 10). These results indicate that the inhibitory effect of NPS1034 combined with osimertinib is not mediated by NF- κ B.

Moreover, we evaluated the effect of an AKT inhibitor (buparlisib). Buparlisib did inhibit the viability to a level similar to that observed following NPS1034 treatment combined with osimertinib.

Collectively, these results suggest that the activation of the AKT-mediated signal may play an important role in the escape of cell death via AXL. We have added these results and revised the statements in the Results (page 14, lines 16-18, page 15, line 1-3, 5), the Discussion (page 22, lines 1-3) and Materials and methods (page 26, lines 6-7) sections.

It would be wonderful if we could evaluate the association between the expression of relevant molecules and osimertinib resistance in clinical specimens. To investigate this issue in our study, we would need paired clinical specimens obtained before and 3-9 days after the initiation of osimertinib treatment, because our study showed that AXL phosphorylation is increased by osimertinib exposure. In fact, constitutive expression of AKT and phosphorylated AKT (pAKT), as well as SPRY4 and ERK, did not correlate with the sensitivity of the *EGFR*-mutated NSCLC cell lines to osimertinib (Supplementary Figures 4 and only for reviewer Figure 3A, 3B).

Moreover, as tumors decrease in size because of osimertinib treatment in the majority of *EGFR*-TKI naïve *EGFR*-mutated NSCLC patients, such re-biopsy may be technically and ethically difficult. Therefore, we could not perform the suggested analysis with paired clinical specimens.

(4) Is the role of AXL specific for osimertinib, or does this relate to other 3rd generation *EGFR* TKIs as well?

Reply:

We examined the effect of the 3rd generation *EGFR*-TKI rociletinib on the viability of *EGFR*-mutated NSCLC cells with or without AXL expression. Similar to osimertinib (Figure 4A), the AXL inhibitor NPS1034, combined with rociletinib, reduced the viability of the high AXL-expressing PC-9, PC-9GXR, and HCC4011 cells, but not that of the low AXL-expressing HCC827 cells. We have included these results in the Results section (page 14, lines 1-7) and showed the data in supplementary Figure 7.

(5) Is the AXL-mediated resistance reversible?

Reply:

To examine whether the AXL-mediated resistance was reversible, we performed additional experiments using PC-9 and HCC4011 cells. First, we induced DT cells by culturing them for 9 days in the presence of 3 μ M osimertinib, as shown in Figure 5A. Then, these cells were cultured in the absence of osimertinib for an additional 9 or 28 days, which was defined as the production of drug-free (DF) cells (DF9 and DF28, respectively). The DF9 cells obtained from the PC-9 and HCC4011 cells showed a similar osimertinib-sensitivity as the DT cells, and the DF28 cells were slightly sensitive compared to the DT or PF9 cells. These results indicated that the AXL-mediated resistance was irreversible at least for 9 days and was not easily reversible (for instance, within 4 weeks). These results are shown as only for reviewer Figure 4.

Reviewer #2 (Remarks to the Author):

1. Results: "However, approximately 20–30% of the PC-9 cells survived, even when treated with a high concentration (10 μ M) of osimertinib for 72 h (Supplemental Figure 1). To determine the mechanism by which these cells escaped the effects of osimertinib,". This is the start point of this study to consider "intrinsic resistance" in a cancer cell population. However, this finding has not been reproduced throughout this study, as presented in Figure 4A, 5A, 6C, and Supplemental Figure 4. Is it really a basis of this study that some PC9 cells escape the effects of osimertinib under this condition?

Reply:

As pointed out correctly, our results showed the efficacy of osimertinib up to 10 μ M. However, the C_{max} in the patient plasma following the oral administration of 80 mg of osimertinib was 635.4 nM (Janne PA et al, NEJM 2015;372:1689-99. We added this paper as ref 25. Accordingly, references were re-numbered in the revised version of manuscript); as such, the 10 μ M concentration of osimertinib used in our *in vitro* experiments was not a clinically achievable concentration. We, therefore, considered that 1 μ M or less was a clinically relevant concentration of osimertinib. Throughout this study, 20–30% of the PC-9 and HCC4011 cells survived after 72 h of treatment with 1 μ M or less of osimertinib. On the other hand, since DT cells were induced by 2 μ M erlotinib or gefitinib in the previous paper (Sharma SV et al, ref 14 in the revised version of manuscript), we induced DT cells by 3 μ M osimertinib in this study.

We have revised the indicated sentence (page 7, lines 5-9), as follows. "As the C_{max} in the patient plasma following the oral administration of 80 mg of osimertinib was 635.4 nM (ref 25), we considered 1 μ M or less as a clinically relevant concentration of osimertinib. However, approximately 20–30% of the PC-9 cells survived, even after treatment with a high concentration (1 μ M) of osimertinib for 72 h (Supplemental Figure 1)."

2. In abstract, the authors say “Activated AXL was associated with EGFR and HER3 in maintaining cell survival and inducing the emergence of cells tolerant to osimertinib”. However, association of AXL with HER3 were not examined in the experiment to see the emergence of DT cells (Figure 5B). In addition, effect of drug-mediated EGFR and AXL co-inhibition on HER3 phosphorylation was examined either. Therefore, the results are not conclusive.

Reply:

We also examined both phosphorylation and total expression of HER3 under drug-mediated EGFR and AXL co-inhibition and found that p-HER3 could be inhibited by osimertinib when combined with NPS1034 in DT cells (Figure 5E), as well as in parental cells, as shown in Figure 4C. These findings are discussed in the Abstract section and were added to the Results section (page 16, lines 8, 13, 14).

3. The experiments using a panel of nine EGFR-mutated lung cancer cell lines in Figures 2 and 3 are interesting and important to generalize the finding of PC9 cell. However, the criteria to divide them into AXL-high and -low cell lines is ambiguous. The authors describe “EGFR phosphorylation tended to be higher in cells with the lower relative AXL expression levels, (page 10)”. The reviewer thinks amounts of EGFR protein rather than phosphorylation show a correlation with the AXL subgroup. How about the expression level of SPRY4? Effect of EGFR and AXL co-inhibition on AKT phosphorylation was not examined in AXL-low cells in Figure 3, so, it is unclear whether this phenomenon is responsible for the differential drug response in Figure 3B or not.

Reply:

As the reviewer has pointed out, the definition of AXL high- and AXL low-expressing tumor cells was unclear in the original version of the manuscript. We performed densitometry of the bands for AXL and β actin in the western blot (Figure 2A) and investigated the correlation between the AXL/ β actin levels and osimertinib sensitivity (IC_{50}) in the *EGFR*-mutated NSCLC cell lines. Importantly, the AXL/ β actin levels strongly correlated with the IC_{50} of osimertinib in these cell lines (Spearman $r = 0.733$; $P = 0.031$), indicating an inverse correlation between AXL expression and osimertinib sensitivity in the *EGFR*-mutated NSCLC cell lines (supplemental Figure 5). Based on these new results, we defined the AXL high-expressing tumor cells by an AXL/ β actin ratio of >7.0 in this study. We added these data to the Results section (page 10, line 17-18, page 11, line 7-9).

Similarly, we next examined the protein correlation between the AXL expression and total or phosphorylated protein expression. The pEGFR levels weakly correlated with the AXL

levels (Spearman $r = -0.6$, $P = 0.048$); whereas, the total EGFR levels did not correlate with the levels of AXL (Spearman $r = -0.233$, $P = 0.276$). These data have been added to the Results section (page 10, line 18, page 11, lines 1-4), supplemental Figure 3A, and only for reviewer Figure 5, respectively.

Moreover, the AXL levels weakly correlated with the SPRY4 levels in the *EGFR*-mutated NSCLC cells (Spearman $r = -0.644$; $P = 0.061$) in supplemental Figure 3B.

We additionally examined the effect of EGFR and AXL co-inhibition on the phosphorylation of AKT and ERK in the AXL low cells HCC827, HCC4006, and H3255. In these cell lines, the AKT phosphorylation was remarkably inhibited by osimertinib alone, and the addition of NPS1034 did not affect the inhibition of AKT phosphorylation. These results have been included in the Results section (page 14, lines 10-12), and the data are shown in supplemental Figure 8.

4. In Table 2C, unfortunately, it is unclear whether osimertinib show the same tendency or not, although most of the experiments in this study have been done with osimertinib.

Reply:

As recommended by the reviewer, we collected new clinical specimens from the *EGFR*-mutated NSCLC patients treated with osimertinib. We could obtain IRB approval for these additional experiments at the four institutes by October 2018. Osimertinib was first approved for T790M-positive *EGFR*-mutated NSCLC patients previously treated with *EGFR*-TKIs in May of 2016, and in August of 2018, it was approved as the first line treatment for patients with untreated *EGFR*-mutated NSCLC in Japan. We could collect 11 tumor specimens from *EGFR*-mutated NSCLC patients at these four institutes and examined the correlation between AXL expression and the outcome after osimertinib treatment in these patients. All specimens were obtained before the initiation of *EGFR*-TKI treatment, and all were positive for the *EGFR*-T790M mutation at the acquisition of resistance to the initial *EGFR*-TKI treatment. No tumor specimens could be collected from the patients who were treated with osimertinib as the first line of treatment, because the clinical response has not been determined at this time point.

In this new set of specimens, high (3+), intermediate (2+), low (1+), and no (0) AXL expression was observed in three (27.2%), one (18.2%), seven (63.6%), and zero (0.0%) specimens, respectively. The response rate for the patients with AXL expression scores of 1+ to 2+ was high (85.7% to 100%), whereas, for the patients with AXL expression scores of 3+, the response rate was relatively lower (66.7%) (Figure 2D). Moreover, the PFS for osimertinib treatment trended toward being shorter in the patients whose AXL expression scores were 3+, compared to those whose AXL expression scores were 0 to 2+ ($P = 0.449$

and HR = 0.81) (Supplemental Figure 6C).

Although the number of patients with an AXL score of 3+ was very limited, these results were consistent with those in shown Figure 2C. Larger scale studies are warranted to further clarify the correlation between AXL expression and the clinical response to osimertinib in EGFR-TKI naïve *EGFR*-mutated NSCLC in the future. We have added these statements in the Results (page 11, lines 10-13, 16-17, 18; page 12, lines 1-12), Discussion (page 24, lines 9, 11-13), and Materials and methods sections (page 31, lines 13-17).

5. Involvement of AXL in acquired TKI resistance has been suggested by several studies as instructed in Introduction. This fact reduces the novelty/significance of this study focusing on intrinsic TKI resistance, since it is conceivable that both mechanisms overlap with each other, as indicated by T790M mutation.

Reply:

While the concept of AXL activation, which is related to EMT features, has already been reported for acquired resistance to the 1st generation EGFR-TKI erlotinib, circumvention of EGFR-TKI resistance is really difficult at the acquired resistance phase in the clinic, except for the circumvention of EGFR-T790M mutation-associated resistance by osimertinib. Therefore, it is worthwhile to determine the novel roles of AXL activation during the initial phase of treatment with the third generation EGFR-TKI osimertinib. In this study, we first revealed the osimertinib tolerant mechanisms via the SPRY4-AXL axis and AXL interaction with other molecules (including EGFR and HER3), independent of EMT features. Second, we demonstrated that treatment during the initial phase with a combination of osimertinib and an AXL inhibitor may prevent the development of intrinsic resistance to osimertinib and the emergence of drug-tolerant cells in *EGFR*-mutated lung cancer overexpressing AXL. Finally, we showed a predictive biomarker to enable the search for poor responders to EGFR-TKI treatment in clinical specimens, before the initiation of EGFR-TKI treatment. Therefore, we believe that this is worthy in the view of novel findings in science, as well as clinical impact.

6. “While AXL was not constitutively phosphorylated, its phosphorylation was induced by osimertinib at 4 h and increased through 72 h (Page 7)... These results suggested that osimertinib exposure may have activated AXL and thereby re-activated HER3, MET, and EGFR in PC-9 cells (Page 8).” These results indicate that AXL is a key trigger for resistance emergence. However, in Supplemental Figure 2A, phosphorylation of AXL is not likely to be induced at 4hr after osimertinib treatment. The reviewer thinks that data in Supplemental Figure 2A is important to explain the resistance mechanism through AXL.

Reply:

We performed a densitometry analysis of the bands for phosphorylated AXL and β actin in the western blot (Supplemental Figure 2A) to quantify the level of phosphorylated AXL. We found that the levels of phosphorylated AXL 4 h after osimertinib exposure had discernibly increased compared to the control, consistent with the results in Figure 1B. We showed these results in only for reviewer Figure 6.

Minor points:

1. Abstract: “TKI” must be spelled out.

Reply:

As suggested, we have spelled it out in the Abstract.

2. The authors should explain the specificity of osimertinib against other kinases than mutated EGFR. How about its effect of the inhibitor on other kinases such as AXL, AKT1 and HER3? Since a variety fields of researchers read Nat Com papers, such a basic explanation is needed.

Reply:

As per the reviewer’s suggestion, we have added the explanation regarding the inhibitory activation by osimertinib in the Introduction section, as follows; “Osimertinib is a third-generation EGFR-TKI that inhibits EGFRs that have activating mutations and/or the T790M resistance mutation, but it does not affect wild-type EGFR or other kinases, such as AXL, AKT1, or HER3.” (page 4, lines 14-15).

3. Page 9: The authors use the term “adversely activated”. The reviewer does not understand why and how the activation is adverse.

Reply:

The text has been revised in accordance with the reviewer’s suggestion.

REVIEWERS' COMMENTS:

Reviewer #1 (Remarks to the Author):

The authors have addressed my questions. The manuscript is suitable for publication. I would recommend including the new figures provided in the rebuttal letter in the final manuscript as supplemental figures, as the data are relevant.

Reviewer #2 (Remarks to the Author):

The revised manuscript has been improved. However, the reviewer feels that the following one point still remains inconclusive.

Table D in Figure 2, analysis of more samples is needed for drawing a conclusion. This data is quite important to address the authenticity of the hypothesis raised by in vitro/in vivo experiments. On the other hand, the reviewer understands that to collect samples needs time.

Point-by-point reply to comments:

Reviewer #1 (Remarks to the Author):

The authors have addressed my questions. The manuscript is suitable for publication. I would recommend including the new figures provided in the rebuttal letter in the final manuscript as supplemental figures, as the data are relevant.

(Reply)

In accordance with the recommendation above, the figures 1A, 1B, 2, 3A, 3B, 4, and 5 from our rebuttal letter have been included as Supplementary Figures 3C, 6, 9A, 9B, 13, 15, and Supplementary Table 1 in the final version of our manuscript.

Further, we have added the grant entitled “Extramural Collaborative Research Grant of Cancer Research Institute, Kanazawa University” given to Tadaaki Yamada and Hirokazu Taniguchi among the Funding Sources (page 40, lines 9-10), since we performed additional experiments described in the aforementioned figures using this grant.

Reviewer #2 (Remarks to the Author):

The revised manuscript has been improved. However, the reviewer feels that the following one point still remains inconclusive.

Table D in Figure 2, analysis of more samples is needed for drawing a conclusion. This data is quite important to address the authenticity of the hypothesis raised by in vitro/in vivo experiments. On the other hand, the reviewer understands that to collect samples needs time.

(Reply)

We agree that studies on a larger scale are warranted to make the data in Table D of Figure 2 more convincing. In fact, we had stated the necessity for performing such studies in the Discussion section of the previous revision of this manuscript (page 24, lines 11-13). However, as acknowledged by the reviewer, it takes a long time to obtain clinical samples from multiple institutes with IRB approval. It was not possible for us to complete such an exercise within the time allotted for this revision (two weeks). Nonetheless, we have included the following statement recognizing this limitation in the manuscript (page 25, lines 13-16): “A limitation of our study is the small number of clinical specimens used in Figure 2D. Larger scale studies are warranted in the future to

further clarify the relationship between AXL expression and the clinical response to osimertinib among EGFR-TKI naïve *EGFR*-mutated NSCLCs.”